# Mineralogical-Petrographical Record of Melt-Rock Interaction and P-T Estimates from the Ozren Massif Ophiolites (Bosnia and Herzegovina)

**Marián Putiš** [1,*], **Ondrej Nemec** [1], **Samir Ustalić** [1], **Elvir Babajić** [2], **Peter Ružička** [1], **Friedrich Koller** [3], **Sergii Kurylo** [4] **and Petar Katanić** [5]

[1] Department of Mineralogy, Petrology and Economic Geology, Faculty of Natural Sciences, Comenius University in Bratislava, 842 15 Bratislava, Slovakia
[2] Department of Mineralogy and Petrology, Faculty of Mining, Geology and Civil Engineering, University of Tuzla, 75 000 Tuzla, Bosnia and Herzegovina
[3] Institute of Lithospheric Research, Faculty of Earth Sciences, Geography and Astronomy, University of Vienna, 1090 Vienna, Austria
[4] Earth Science Institute, Slovak Academy of Sciences, 974 11 Banská Bystrica, Slovakia
[5] Gim Geotehnika d.o.o. Banja Luka, 78 000 Banja Luka, Bosnia and Herzegovina
[*] Correspondence: marian.putis@uniba.sk

**Abstract:** The Dinaride Ophiolite Belt formed from the Jurassic part of the Neotethys. The investigated Ozren ophiolite complex in Bosnia and Herzegovina consists of peridotites, plagioclase peridotites, plagiogranites, troctolites and other gabbroic rocks, and fewer basalts. Lherzolites and harzburgites contain corroded ortho- and clinopyroxene1 porphyroclasts enclosed in the olivine matrix. The boundaries between olivine aggregates and pyroxene1 and spinel1 are infilled by medium-grained undeformed aggregates of clinopyroxene2, less orthopyroxene2, spinel2, and often clinopyroxene3-spinel3 symplectites. These textures indicate the final crystallization of peridotite in subsolidus conditions. Partial dissolution of deformed pyroxene1 porphyroclasts and coarse-grained spinel1 most likely occurred due to their reaction with the rest melt present in the grain boundaries. The Al decrease from pyroxene1 to pyroxene2 and 3, or the Cr decrease and Al increase from spinel1 to spinel2 and 3 is characteristic. Peridotites are associated with inferred remnants of a gabbro-dolerite layer, whereas basalts and radiolarites occur as rare dm-size fragments in an ophiolitic breccia. Troctolites display interstitial crystallization of plagioclase, clinopyroxene, less Na-Ti-rich amphiboles, and phlogopite in the olivine-spinel matrix, indicating the replacive character of impregnating melt within the dunite layers. Clinopyroxene-plagioclase-ilmenite-±amphibole gabbroic and fewer basaltic dykes in peridotites formed due to subridge extension, mantle thinning, and the deeper mantle melting. Iron-enriched olivines occur in the peridotite-dyke interfaces and troctolites. Hydrated ultramafics and mafics contain amphiboles, biotite, phlogopite, clinozoisite, epidote, and chlorite aggregates. Estimated magmatic to subsolidus T from peridotite two-pyroxene thermometry are 1000–850 °C, for the spinel facies. Ca-in-orthopyroxene1 thermometry provided T of 1028–1068 °C, and Ca-in-orthopyroxene2 thermometry gave 909–961 °C at estimated P of 1.1–0.9 GPa. However, the gabbroic dyke magmatic crystallization T was constrained to 1200–1100 °C at P of 0.45–0.15 GPa by single clinopyroxene thermobarometry. The obtained P–T conditions constrained the deeper mantle environment for the formation of peridotites than troctolites and cross-cutting dykes. The ophiolitic thrust-sheet hanging wall conditions in an obduction-related accretionary wedge were estimated from amphibolites at 620 °C and 0.85 GPa by Ti-in-amphibole thermometry and amphibole-plagioclase thermobarometry. 300 °C and 0.5 GPa were determined from an exhumation shear zone using a combination of chlorite thermometry and Si-in-phengite barometry.

**Keywords:** petrography; mineral chemistry; melt-rock interaction; P–T estimates; Ozren ophiolites; Bosnia and Herzegovina

## 1. Introduction

The mantle rocks at extensional settings record varying degrees and depths of partial melting and crystallization, leaving refractory peridotites (e.g., [1]). The chemical modifications of peridotites are moreover caused by melt-rock interaction due to buoyant percolating olivine (Ol)-saturated melt in the asthenospheric thermal boundary layer (e.g., [2–7]) at temperatures preventing crystallization. However, continuing mantle lithosphere thinning may be accompanied by additional fractionated melt percolation through peridotites, and thus a newer period of the melt-rock interaction can occur at a shallower mantle level. Consequently, plagioclase (Pl)-rich peridotites have been interpreted as the replacive product of melt impregnation at shallower Pl-facies conditions, leading to Ol dissolution and interstitial Pl and pyroxene (Px) crystallization (e.g., [8–13]).

An example of the above-mentioned scenarios was reported from the Erro-Tobbio ophiolitic unit in Italy, which consists of variably serpentinized spinel-bearing lherzolites and harzburgites. They represent the Jurassic Ligurian Tethys oceanic basin opened by passive lithosphere extension and breakup of the continental lithosphere leading to slow-spreading oceanization [14,15]. The extension-related exhumation there was accompanied by multiple episodes of melt percolation and intrusion: (1) The first open-system Ol-saturated reactive porous flow [16] at spinel (Spl)-facies conditions (1.5–2 GPa) leading to the dissolution of mantle clinopyroxene (Cpx) and orthopyroxene (Opx) and crystallization of Ol in reactive peridotites to replacive dunites; (2) a melt–rock reaction at Pl-facies conditions (<0.8–1 GPa) leading to the formation of Pl-bearing impregnated peridotites by dissolution of Ol and crystallization of Pl ± Opx, ± Cpx; and (3) multiple episodes of gabbroic intrusions at the shallower mantle depths (<0.5 GPa) [6,9,16–19].

Our field investigation of the Ozren ophiolites revealed the dunite layers in peridotites, which may have formed by the reactive porous flow processes, a higher melting degree, and/or a complete basaltic melt extraction. Microstructures of peridotites contain Ol aggregates with the deformed Opx1 and Cpx1 porphyroclasts and Spl1, with the latter showing partial dissolution. The Ol-Opx1-Cpx1-Spl1 boundaries are infilled with the melt crystallized undeformed Cpx2 and fewer Opx2 and Spl2 aggregates, and are additionally accompanied by the Cpx3-Spl3 symplectites. The question has arisen whether these boundary aggregates and symplectites formed from a Ca-Si-Al enriched rest melt due to an incomplete melt extraction, or do they already represent a newer percolating melt yet in the Spl stability field?

Next, we focused on the impregnated Pl-bearing peridotites with troctolitic and other gabbroic layers or crosscutting dykes and veins that were formed by the mechanism of melt impregnation, often called "refertilization" [20,21].

This study reports spatial and textural relationships between the Spl peridotites (lherzolites, harzburgites, rare dunites) and the Pl-Cpx-(±amphibole-Amp, phlogopite-Phl) percolation domains through the peridotites. There are three different cases of the melt-rock interaction reported from our samples: (1) The formation of reactive peridotites, including dunites, in the deeper mantle lithosphere, (2) transformation of dunite layers into troctolites, and (3) the interaction between peridotites and crosscutting gabbroic dykes and veins, with (2) and (3) occurring in the shallower mantle lithosphere environment due to lithosphere extension and thinning. We provide characterization of magmatic and post-magmatic mineral assemblages and preliminary P–T estimates for the magmatic formation of ultramafic and mafic rocks in the subridge environment, and subsequent burial and exhumation in an obduction-related accretionary wedge.

The obtained results are supported by field data such as mesostructural measurements, lithological transsections, and a borehole core profile. The laboratory study included polarized light (PL) microscopy of polished thin sections with an emphasis on microstructures, modal analyses, and electron probe micro-analysis (EPMA) of the representative rock-forming minerals, with the latter also used for the P–T estimates of the magmatic crystallization and the metamorphic overprint in an accretionary wedge.

Abbreviations of the rock-forming mineral names used in the text, tables, and figures are as follows [22]: Ab, albite; Act, actinolite; Amp, amphibole; An, anorthite; Ap, apatite; Aug, augite; Bt, biotite; Chl, chlorite; Cpx, clinopyroxene; Czo, clinozoisite; Di, diopside; Ep, epidote; Fkrs, ferro-kaersutite; Fsp, feldspar; Grt, garnet; Gru, grunerite; H-Grs, hydro-grossular; Hbl, hornblende; Hst, hastingsite; Ilm, ilmenite; Krs, kaersutite; Ktp, katophorite, Mag, magnetite; Mhb, magnesio-hornblende; Ms, muscovite; Ol, olivine; Opx, orthopyroxene; Phg, phengite (~Cel-rich Ms); Phl, phlogopite; Pl, plagioclase; Prg, pargasite; Px, pyroxene; Qz, quartz; Rt, rutile; Sdg, sadanagaite; Spl, spinel; Srp, serpentine (group); Str, strontianite; Tr, tremolite; Ttn, titanite; Wnc, winchite; Zo, zoisite; Zrn, zircon.

## 2. Geological Settings

The Dinaridic segment of Neotethys was affected by widespread shortening and related subduction–accretion–obduction processes that commenced in the Middle Jurassic ([23–27], and references therein). The Dinaride ophiolites belong either to the Central Dinaridic Ophiolite Belt (CDOB) or to the Vardar zone (VZ, also called IDOB = Inner Dinaridic Ophiolite Belt). The main differences are (a) diverse genetic models of origin, (b) different ages of the formation, and (c) different structures, especially concerning their relations to underlying and overlying sedimentary sequences. Dismembered CDOB ophiolites are related to the ocean-ridge geotectonic setting, whereas patches of the VZ ophiolites are of supra-subduction origin (e.g., [28]).

The CDOB and VZ Western Belt [24,29,30] are collectively referred to as the Western Vardar Ophiolite Unit by [31] to distinguish them from the ophiolites of the Eastern Vardar Ophiolite Unit (or Main Vardar Belt, [30]), separated from the Western Unit by the Sava Zone [31].

These ophiolite complexes (OC) can be distinguished in the CDOB: 1. Kozara OC, 2. Ljubić and Čavka OC, 3. Borja and Mahnjača OC, 4. Ozren OC (OOC), 5. Krivaja-Konjuh OC (KKOC), 6a. Varda (Višegrad) OC, 6b. Zlatibor OC, and 7. Ozren (Sjenica) OC, which are depicted in Figure 1.

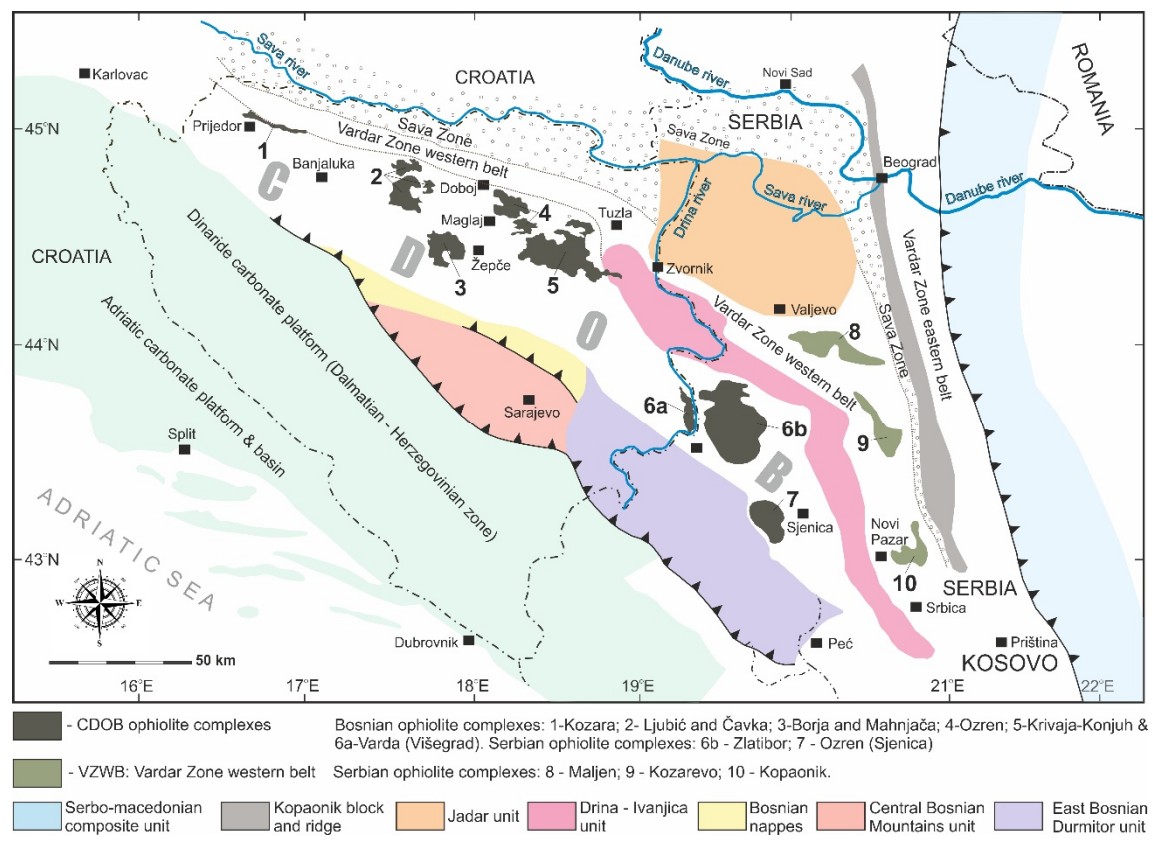

**Figure 1.** Ozren Massif (4) in Central Dinaridic Ophiolite Belt (after [31,32], modified).

From the known ophiolite complexes of the Dinarides, the OOC (Figures 2 and 3, with the sample location) is the second largest complex occupying an area of approximately 300 km². The first geological investigations on the CDOB began in the late 19th century but the first, most detailed research of this complex was conducted by [33]. This research and all the following investigations had a predominantly regional-geological character and revealed the basic structure of the OOC composed of serpentinized peridotites, different types of gabbros, fewer basalts, radiolarites, and amphibolites. The ophiolitic mélange is crosscut by inferred Neogene andesites and dacites and is covered by Neogene sediments (Figure 2).

Recent studies were mainly focused on the neighboring and largest Krivaja-Konjuh ophiolite complex (KKOC) by [26–28,34]. The Krivaja-Konjuh Massif consists of Spl and Pl peridotites, gabbros (including troctolitic), and fewer basalts. The peridotites mainly consist of lherzolites with Cpx contents from 5% to 15%. Dunites and harzburgites are rare, and no signs of prograde metamorphism were observed in peridotites or gabbros by [28] and [35]. However, the chromitite occurrences may suggest their subduction-related origin, indicating that the massif became a part of the upper plate during closure of the ocean basin. Basaltic dykes in the south-eastern part of the massif have supra-subduction major and trace element affinity, indicating at least a short-lived intra-oceanic subduction in the KKOC [36]. The high Cpx $Na_2O$ contents suggest a sub-continental origin of parts of Krivaja peridotites, while the Spl peridotites in Konjuh have significantly lower Cpx $Na_2O$ contents, typical of sub-oceanic environment [28]. Spinel peridotites from KK do not range to the depleted end of abyssal harzburgites, with Cpx $Al_2O_3$ contents as low as 2% and $TiO_2$ below 0.1% (e.g., from the Mid-Atlantic Ridge [37]). Previous authors [28] reported that the depleted abyssal peridotite compositions partly overlap with the Eastern Mirdita ophiolite, which is inferred to originate in a supra-subduction setting [38,39]. Thus, the compositional characteristics of Krivaja indicate a sub-continental origin, while

those of Konjuh suggest a sub-oceanic origin, although both parts are similar in their fertility [28].

The Vardar Zone massifs are at the depleted end of abyssal peridotites, possibly transitioning to supra-subduction affinity. The gradual change in compositions from west to east with a fertile western continental margin and a depleted, possibly subduction-influenced margin on the eastern side is consistent with the 'single ocean' hypothesis of the peridotite massifs of the Dinarides and Vardar zone of [31].

The CDOB contains amphibolitic to granulitic soles of the thrust sheets (e.g., [40–44]). Metamorphic sole rocks of the KKOC are reported to form an elongated domain between the two main Krivaja and Konjuh ultramafic masses. The peak temperature and pressure conditions calculated from different mineral pairs were estimated between 850 and 1100 °C at 1.1 to 1.3 GPa [27]. The Sm-Nd isochrone age, calculated from Cpx, Pl, Grt, Amp, and whole rock of the sole to 162±14 Ma, was interpreted to date the intra-oceanic subduction within the KKOC [34].

There have been numerous complex or review works published on the CDOB, which may be suitable for the correlation of the OOC with other ophiolitic complexes of this large Neotethyan Jurassic ophiolitic belt, e.g., [23–36].

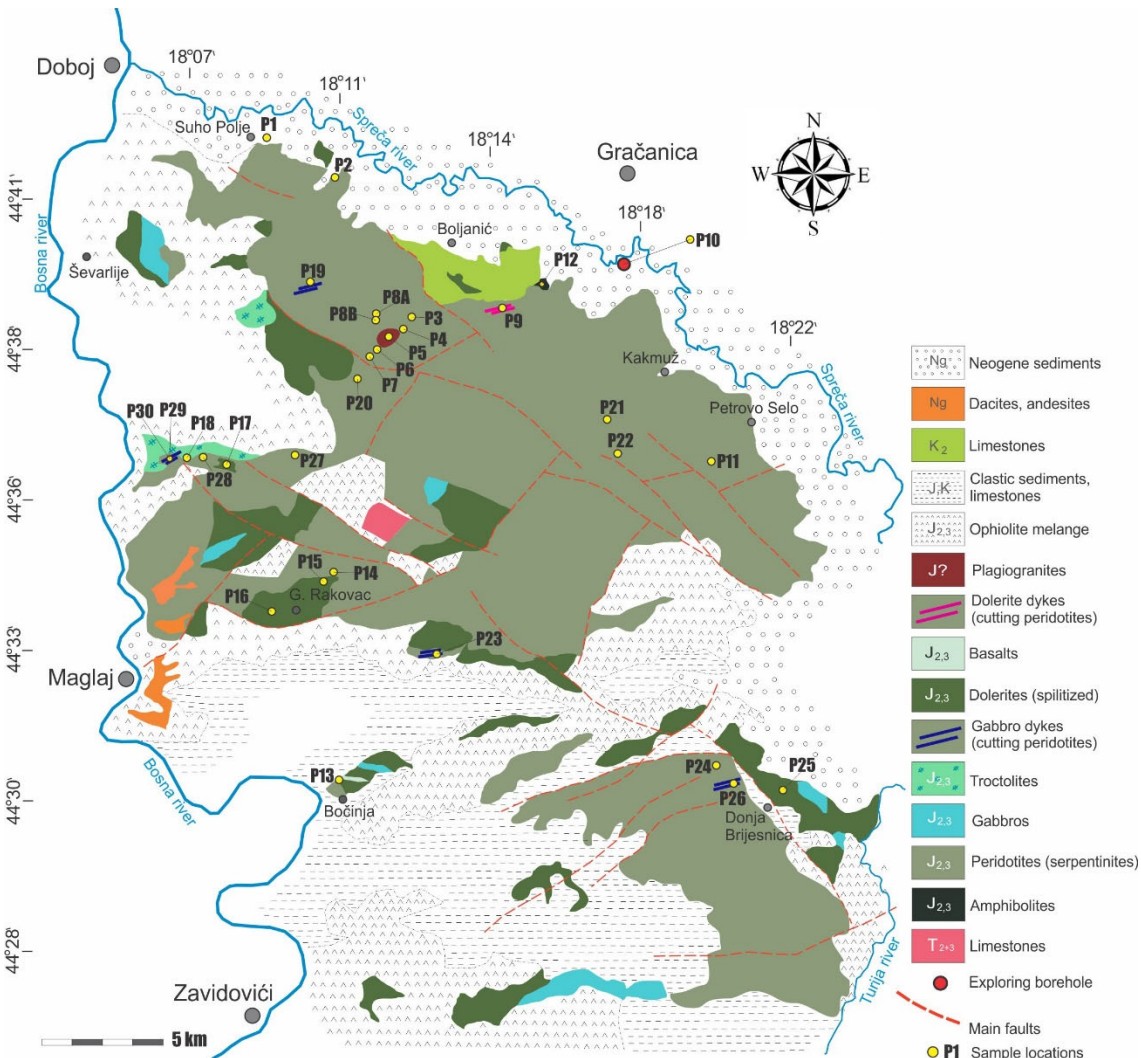

**Figure 2.** Ozren ophiolitic complex. Geological map compiled from [33], two geological sheets (Zavidovići, Doboj) of the basic geological map of the former SFR Yugoslavia 1:100,000 [45,46], and new data from our investigation with sample numbers.

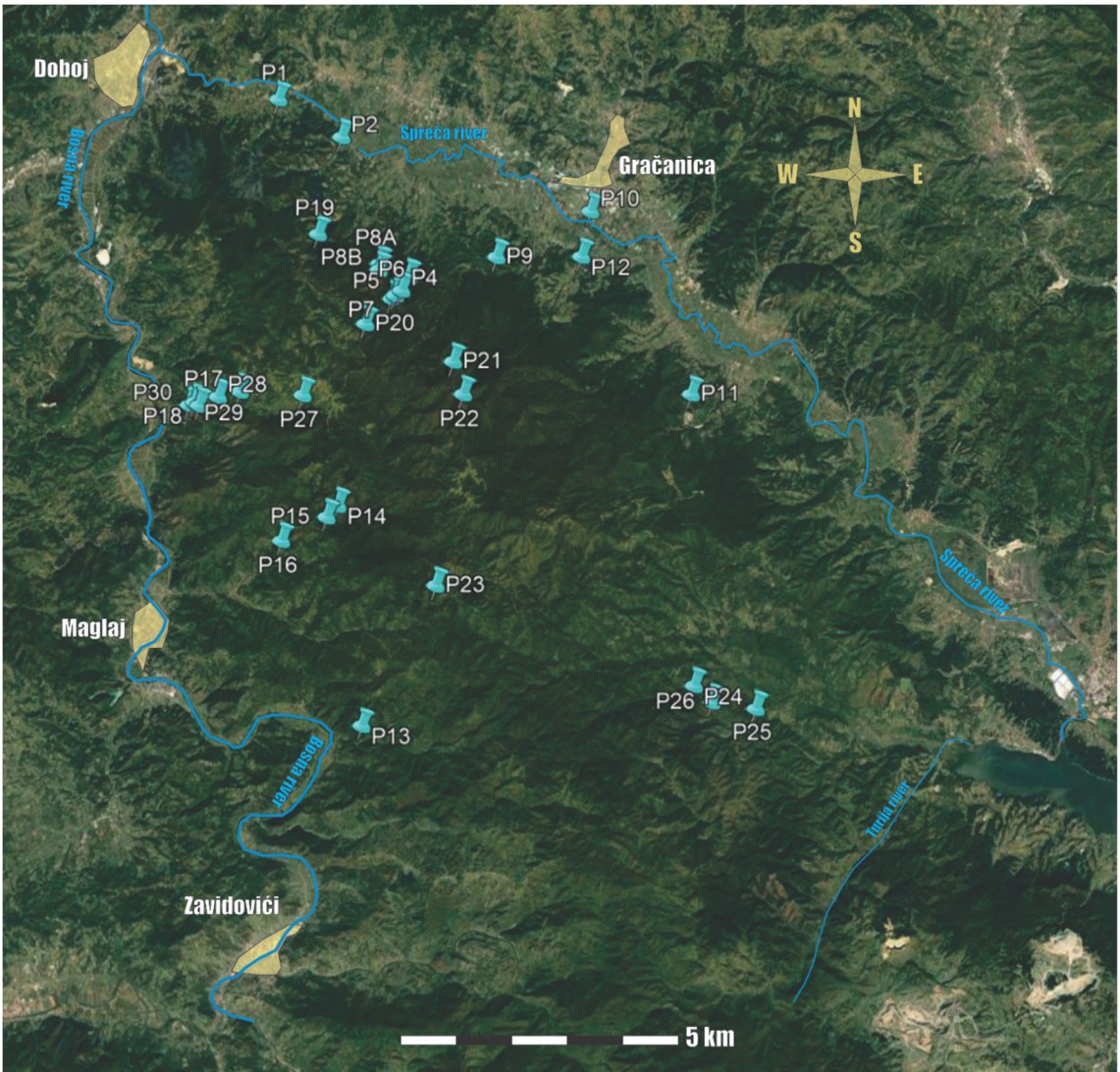

**Figure 3.** Sample location in the Ozren ophiolitic complex [47].

## 3. Materials and Methods

The aim of this detailed petrographical and mineralogical study is to reveal the rock composition of the OOC and utilize the mineral chemistry to constrain the magmatic and post-magmatic processes, providing a database for future geochemical, isotopic, and geochronological study.

The field investigation was focused on finding relatively fresh ultramafic and mafic rocks, suitable for further laboratory investigation (see GPS of the sample location in Table 1 and Figures 2 and 3). All principal rock types were subjected first to transmitted light PL microscopy from polished thin sections. Representative ultramafics were used for modal analysis by visual percentage estimation of the rock-forming minerals.

The mineral element compositions were measured by electron probe micro-analysis (EPMA) on a Cameca SX-100 electron microprobe at the State Geological Institute of Dionýz Štúr in Bratislava, and by the JEOL Super-probe JXA-8530F at the Earth Science Institute of Slovak Academy of Sciences in Banská Bystrica, Slovakia (more than 600 analyses). The electron beam accelerating potential was 15 kV, with a beam current of 20 nA and a beam diameter of 3–5 μm. Detection limits were within 0.01–0.05 wt.% of oxide.

We chose representative rocks and their mineral chemistry for the first P–T estimates of the magmatic and metamorphic evolutionary stages of the OOC. These include Cpx-Opx and Ca-in-Opx thermometry [48] for ultramafics, and single Cpx thermobarometry

for mafics [49]. To constrain metamorphic conditions in an accretionary wedge, the Ti-in-Amp thermometry [50] in combination with Amp-Pl thermobarometry [51] and Chl thermometry [52] in combination with Si-in-Phg barometry [53] were applied.

**Table 1.** GPS locations of investigated samples (22 ultramafics, 33 mafics, 1 radiolarite).

| Samples | GPS Coordinates (N, E) |
|---------|------------------------|
| P1-dolerite altered | N 44°42′46.50″, E 18°9′12.78″ |
| P2-lherzolite | N 44°42′2.64″, E 18°10′55.68″ |
| P3-serpentinite | N 44°39′20.82″, E 18°12′45.06″ |
| P4-serpentinite | N 44°39′5.34″, E 18°12′33.78″ |
| P5A,B-plagiogranite | N 44°39′1.80″, E 18°12′25.32″ |
| P6-lherzolite | N 44°38′59.68″, E 18°12′23.52″ |
| P7-lherzolite | N 44°38′55.62″, E 18°12′17.64″ |
| P8A-Pl lherzolite | N 44°39′36.06″, E 18°11′57.48″ |
| P8B-lherzolite | N 44°39′29.04″, E 18°11′56.70″ |
| P9-dolerite dyke in peridotite | N 44°39′43.82″, E 18°15′7.98″ |
| P10A-C-gabbro from borehole | N 44°40′36.61″, E 18°17′40.59″ |
| P10D-basalt dyke in gabbro from borehole | N 44°40′36.61″, E 18°17′40.59″ |
| P11A-dunite layer in peridotite | N 44°37′5.40″, E 18°20′24.06″ |
| P11B-harzburgite | N 44°37′5.40″, E 18°20′24.06″ |
| P12A-amphibolite | N 44°39′44.33″, E 18°17′25.70″ |
| P12B-amphibolite from borehole | from a borehole core |
| P13A,B1-gabbro from breccia | N 44°30′40.86″, E 18°11′29.58″ |
| P13B2-leucogabbro vein in gabbro from breccia | N 44°30′40.86″, E 18°11′29.58″ |
| P13C,D-dolerite from breccia | N 44°30′42.51″, E 18°11′25.26″ |
| P13E-radiolarite from breccia | N 44°30′40.86″, E 18°11′29.58″ |
| P13F-basalt from breccia | N 44°30′40.86″, E 18°11′29.58″ |
| P14-serpentinite | N 44°34′56.46″, E 18°10′51.00″ |
| P15-dolerite | N 44°34′43.20″, E 18°10′33.00″ |
| P16-dolerite | N 44°34′15.72″, E 18°9′18.84″ |
| P17A-C-dolerite dyke in peridotite | N 44°37′8.26″, E 18°8′11.46″ |
| P18A-C-troctolite in dunite | N 44°36′52.06″, E 18°7′4.42″ |
| P18D-dunite | N 44°36′52.06″, E 18°7′4.42″ |
| P18G2-dolerite dyke in troctolite | N 44°36′52.06″, E 18°7′4.42″ |
| P19A1-gabbro dyke in dunite | N 44°40′10.50″, E 18°10′17.88″ |
| P19A2-troctolite in dunite | N 44°40′10.50″, E 18°10′17.88″ |
| P20-serpentinite | N 44°38′25.98″, E 18°11′35.82″ |
| P21-harzburgite | N 44°37′42.00″, E 18°13′58.86″ |
| P22-serpentinite | N 44°37′4.68″, E 18°14′14.64″ |
| P23A-gabbro dyke mylonite in serpentinite | N 44°33′24.54″, E 18°13′29.94″ |
| P24-serpentinite | N 44°31′28.50″, E 18°20′28.26″ |
| P25-dolerite | N 44°31′1.62″, E 18°22′8.52″ |
| P26A,B-gabbro dyke altered in serpentinite | N 44°31′9.54″, E 18°20′57.06″ |
| P26C-E-serpentinite | N 44°31′9.54″, E 18°20′57.06″ |
| P27-serpentinite | N 44°37′4.02″, E 18°9′54.36″ |
| P28-serpentinite | N 44°37′1.91″, E 18°7′35.66″ |
| P29A1-gabbro dyke in harzburgite | N 44°36′56.04″, E 18°6′56.94″ |
| P29A2-harzburgite | N 44°36′56.04″, E 18°6′56.94″ |
| P29C1-gabbro dyke in harzburgite | N 44°36′55.23″, E 18°6′52.96″ |
| P29C2-harzburgite | N 44°36′55.23″, E 18°6′52.96″ |
| P30-gabbro dyke altered in serpentinite | N 44°36′52.56″, E 18°6′47.70″ |

## 4. Results

### 4.1. Petrography of Peridotites

Peridotites are lherzolites, harzburgites, and rare dunites (Figures 4–8). Here, we describe the main macro- and microscopic features of these rocks—the structure and mineral composition.

#### 4.1.1. Lherzolites

Lherzolites build the Gostilj Peak area in the northern part of the Ozren Massif (samples P2, P6, P7, P8A).

Lherzolites contain Ol (55%), Opx1 (25%), Cpx1 (10%), and Spl1 (1%). The rest is composed of tiny Px and Spl aggregates. Most samples display anisotropic mesostructures such as layering (Figure 4A–C). Microscopic anisotropy is defined by flattened Opx and Cpx porphyroclasts in a partly serpentinized Ol matrix (Figure 4D,E).

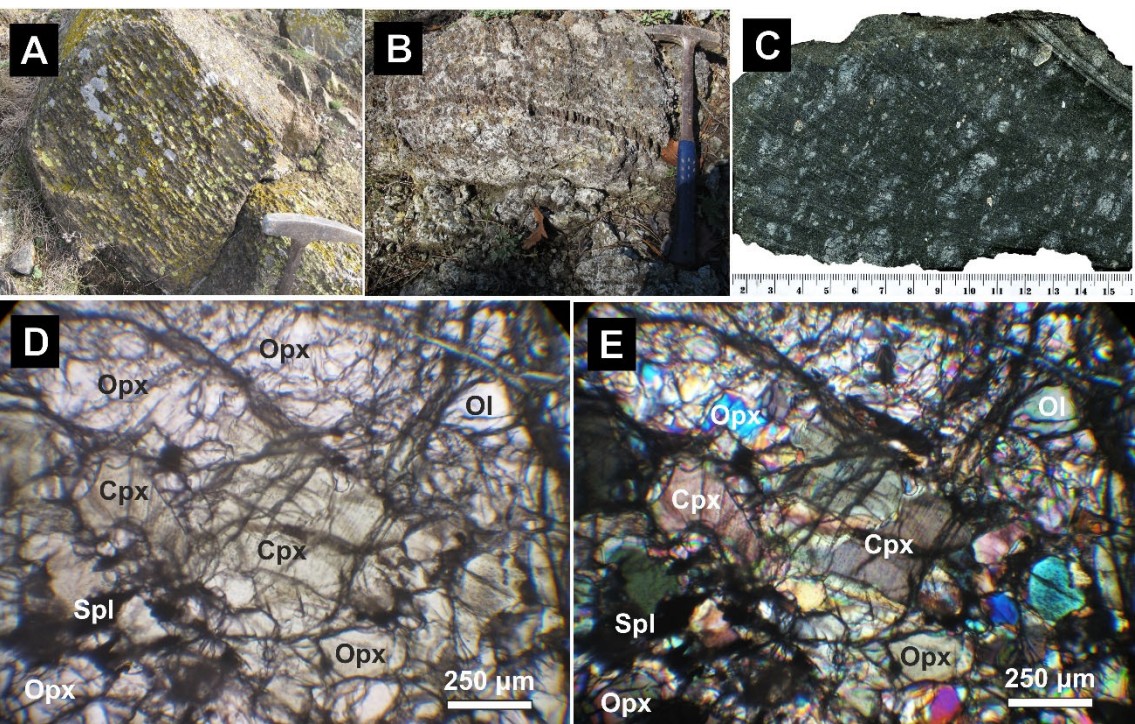

**Figure 4.** Macroscopic and PL microscopic images of lherzolites (A,D,E–s. P8A; B–s. P7; C–s. P2). (**A,B**) Macroscopic layering by alternation of softer serpentinized Ol-rich layers with Px-rich ones. (**C**) Cut surface with pale Px-rich stripes. (**D,E**) Cpx aggregates separating flattened Opx porphyroclasts in slightly serpentinized Ol matrix. Picture D at *II* P; E at *X* P.

The BSE images (Figure 5A–F) reveal the lherzolite structure in more detail. Large Spl1 grains usually occur in the Ol matrix (Figure 5A). Orthopyroxene (Opx1) porphyroclasts have Cpx exsolution lamellae, and similarly, porphyroclastic Cpx1 shows Opx exsolution lamellae (Figure 5B,C). The late magmatic Cpx2, Opx2, and Spl2 aggregates grow in the Opx1-Ol boundary or the Ol matrix (Figure 5B–F). This Px generation may also occur as the undeformed rim of the deformed (kinked, bent) Opx1 and Cpx1 porphyroclasts and does not contain the exsolutions (Figure 5B,F). The latest magmatic Opx2-3–Spl2-3 and Cpx3-Spl3 symplectites formed in the same domains (Figure 5B,E,F). Tiny Spl(3) and Mag are also scattered in the Ol matrix.

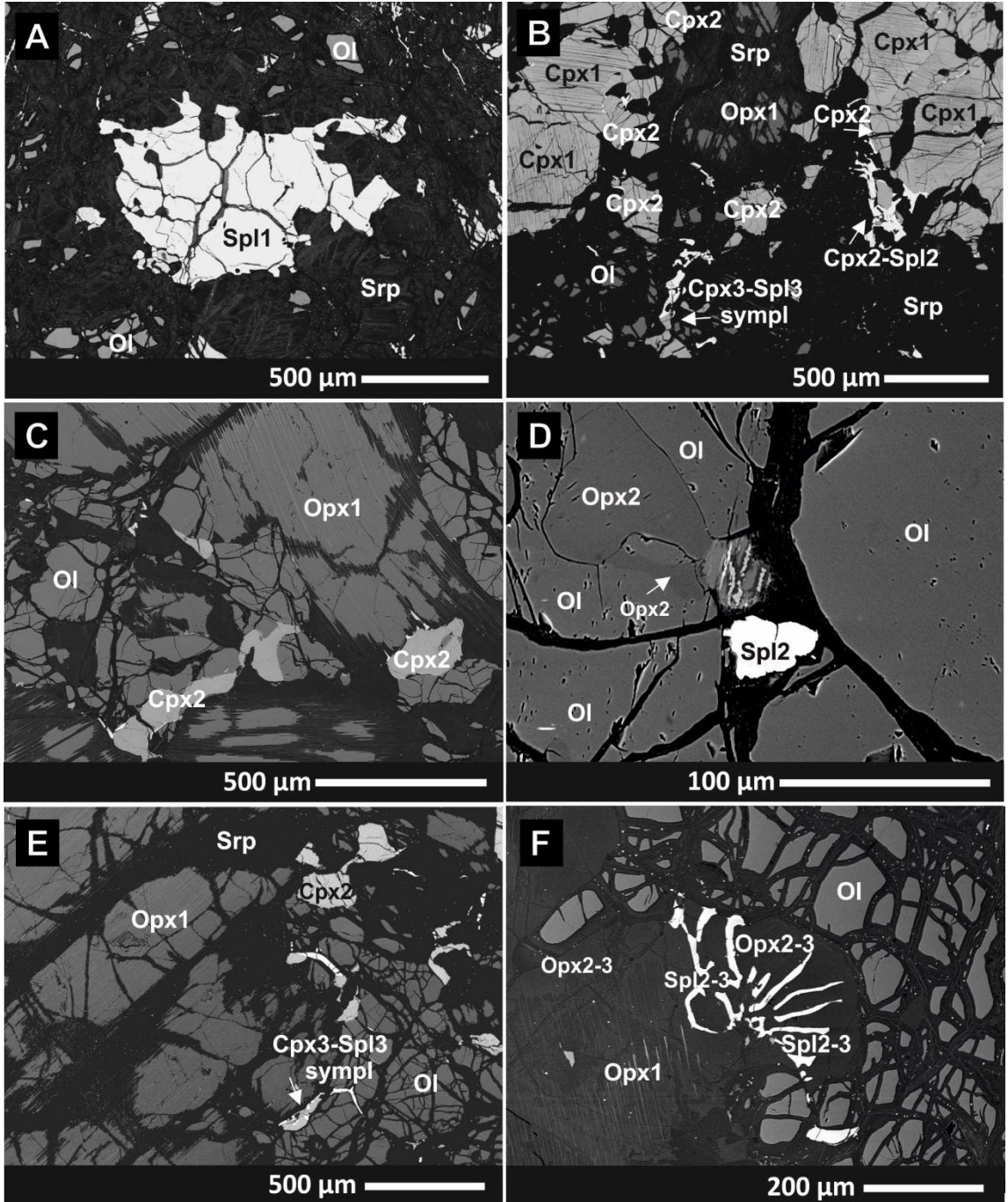

**Figure 5.** BSE images of lherzolites. (**A**) Spl1 in serpentinized Ol matrix (P2). (**B**) Cpx1 porphyroclastic aggregates with Opx exsolutions. Cpx1 has a rim of Cpx2 without exsolutions. Cpx2-Spl2 aggregates and Cpx3-Spl3 symplectites (P2). (**C,D**) Opx1 porphyroclasts with Cpx exsolutions. Opx2, Cpx2, and Spl2 aggregates follow Ol-Opx1 and Ol-Cpx1 boundaries (P2, P8). (**E**) Cpx3-Spl3 symplectites (P2) at Ol-Opx1 boundary and in the Ol matrix. (**F**) Detailed view on Opx1 rim with Opx2-3–Spl2-3 symplectites (P2).

### 4.1.2. Harzburgites

Harzburgites build, e.g., the highest peak of the Ozren Massif—Velika Ostravica (sample P21). The southern part of the massif, approximately S of the Petrovo Selo–Donja Paklenica tectonic (?) line, is mostly built of harzburgites (P11B, P21, P29A2, P29C2), often showing flattened and stretched Opx porphyroclasts in the serpentinitic matrix (Figure 6A).

Olivine predominates in the serpentinized matrix (60%). They contain approximately 35% Opx and less than 5% Cpx. Anhedral Opx porphyroclasts have a grain size of 1–10 mm. Clinopyroxene grows around Opx or within the Ol matrix, but it often follows Ol-Opx boundaries. Clinopyroxene exsolution lamellae in deformed Opx porphyroclasts are characteristic. Spinel occurs in the form of anhedral grains (Figure 6B,C).

The BSE images (Figure 7A–D) reveal the harzburgite structure in more detail. Orthopyroxene (Opx1) porphyroclasts have Cpx exsolution lamellae, and similarly, porphyroclastic Cpx1 has Opx exsolution lamellae (Figure 7A–C). Late magmatic Cpx2 follows the Ol-Opx1 boundary and often extends into the Ol matrix. This Px generation usually does not contain exsolutions (Figure 7A,D). The Cpx2 is accompanied by tiny (<100μm) Spl2 grains (Figure 7A–D).

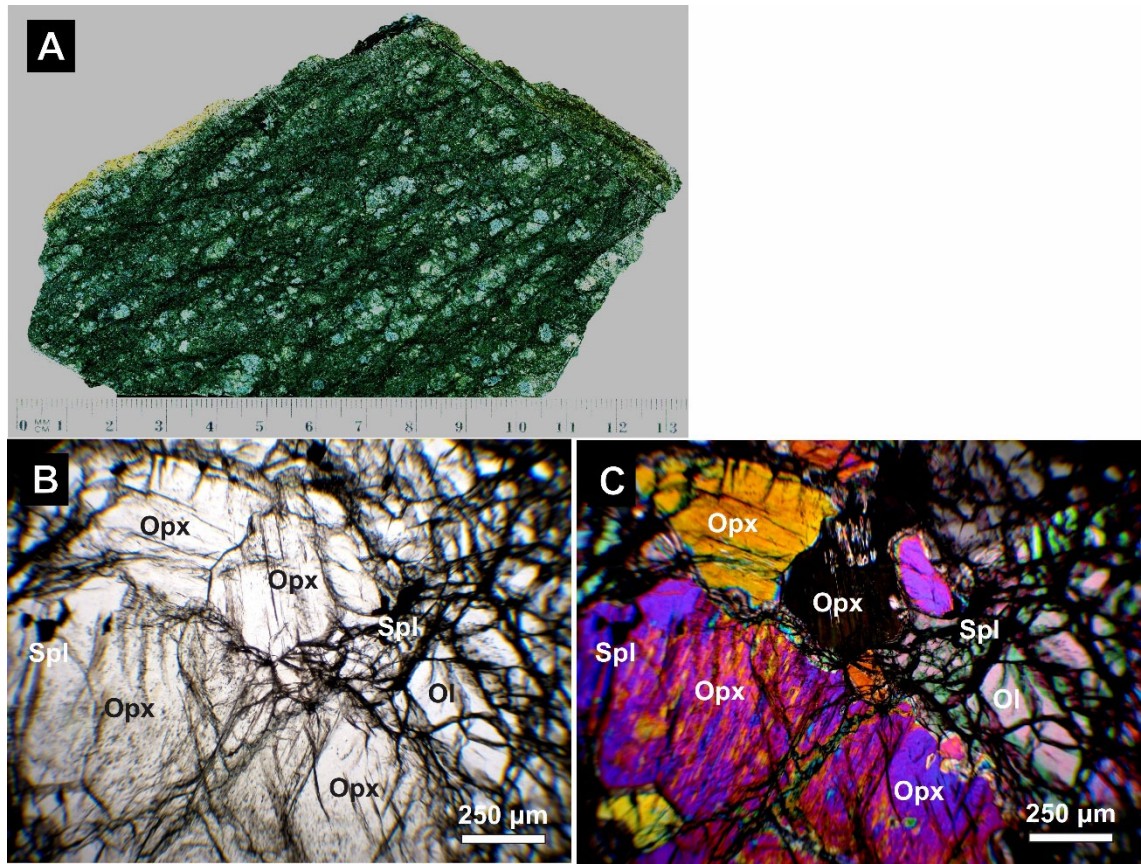

**Figure 6.** Macroscopic and PL microscopic images of harzburgite (P21). (**A**) Cut surface shows macroscopic anisotropy defined by Px porphyroclasts in soft, serpentinized Ol-rich matrix. (**B,C**) Opx porphyroclasts with exsolutions in partly recrystallized and serpentinized Ol matrix. Picture A at *II* P; B at *X* P.

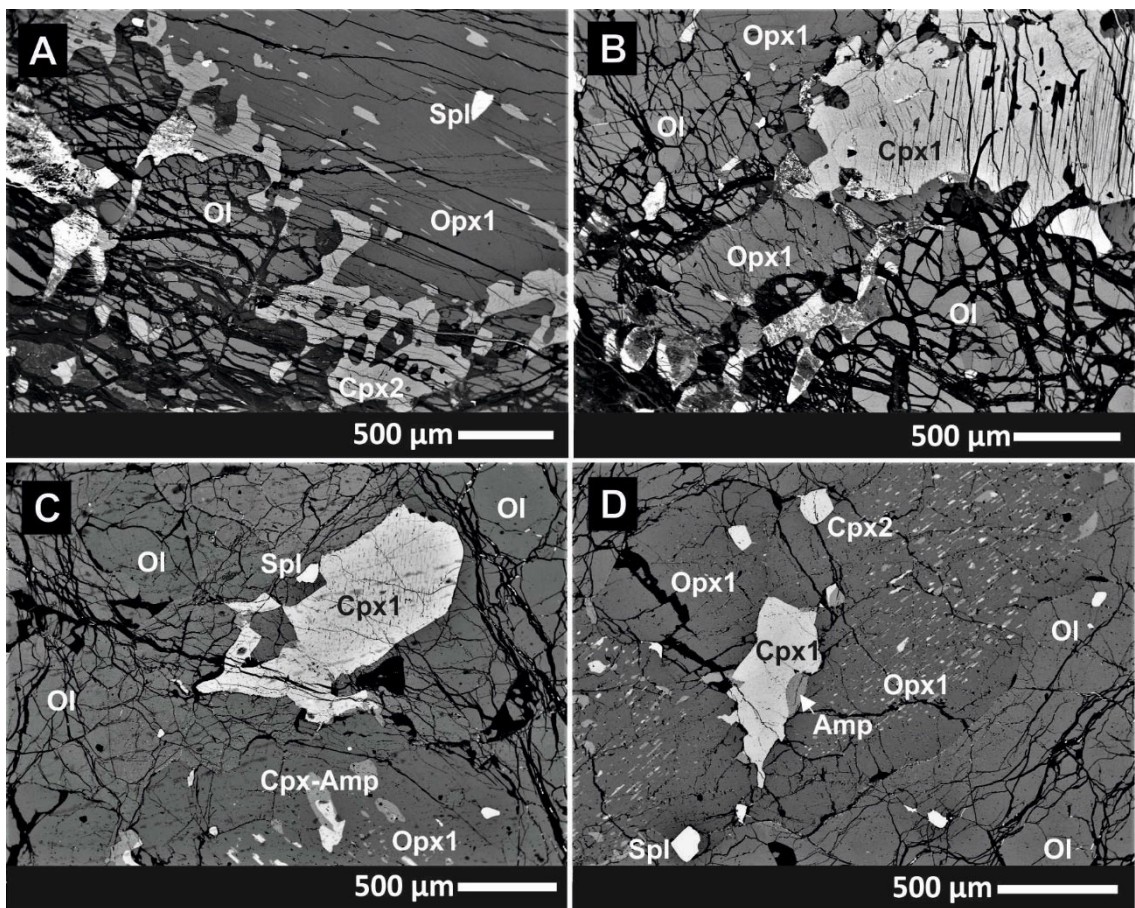

**Figure 7.** BSE images of harzburgites document magmatic to subsolidus stage mineral generations. (**A,B**) Partly dissolved Opx1 and Cpx1 porphyroclasts occur in the Ol matrix. Opx1 contains Cpx exsolutions, and Cpx1 shows Opx and Cr-Spl exsolutions. Cpx2 follows Px1 boundaries and is partly extending into the Ol matrix (P29A2). (**C,D**) Opx1 contains Cpx exsolutions partly replaced by Amp-Mhb. Cpx2 and Spl2 anhedral grains in boundaries of Px1 and the Ol matrix (P21).

### 4.1.3. Dunites

Dunites are rare rock types of the OOC. So far, we found only three dunite layers in harzburgite. Sample P19A2 from the northern part of the massif is, in fact, troctolite with relic dark pods of dunite (Figure 8A). Sample P11A (Figure 8B–E) is from a peridotite quarry S of the Petrovo Selo village on the eastern side of the massif, and sample 18D (Figure 8F) is from the Donja Paklenica Valley at the western side of the OOC.

The massive macroscopic character of these rocks reflects a homogeneous texture composed of serpentinized Ol and accessory Spl (Figure 8C–F). Sample P18D (Figure 8F) was collected from a contact of Pl harzburgite with troctolite, and similarly, troctolite sample P19A2 (Figure 8A) formed along the intrusion contact of a gabbroic dyke into a dunite layer in peridotite (see the part 4.3.2.).

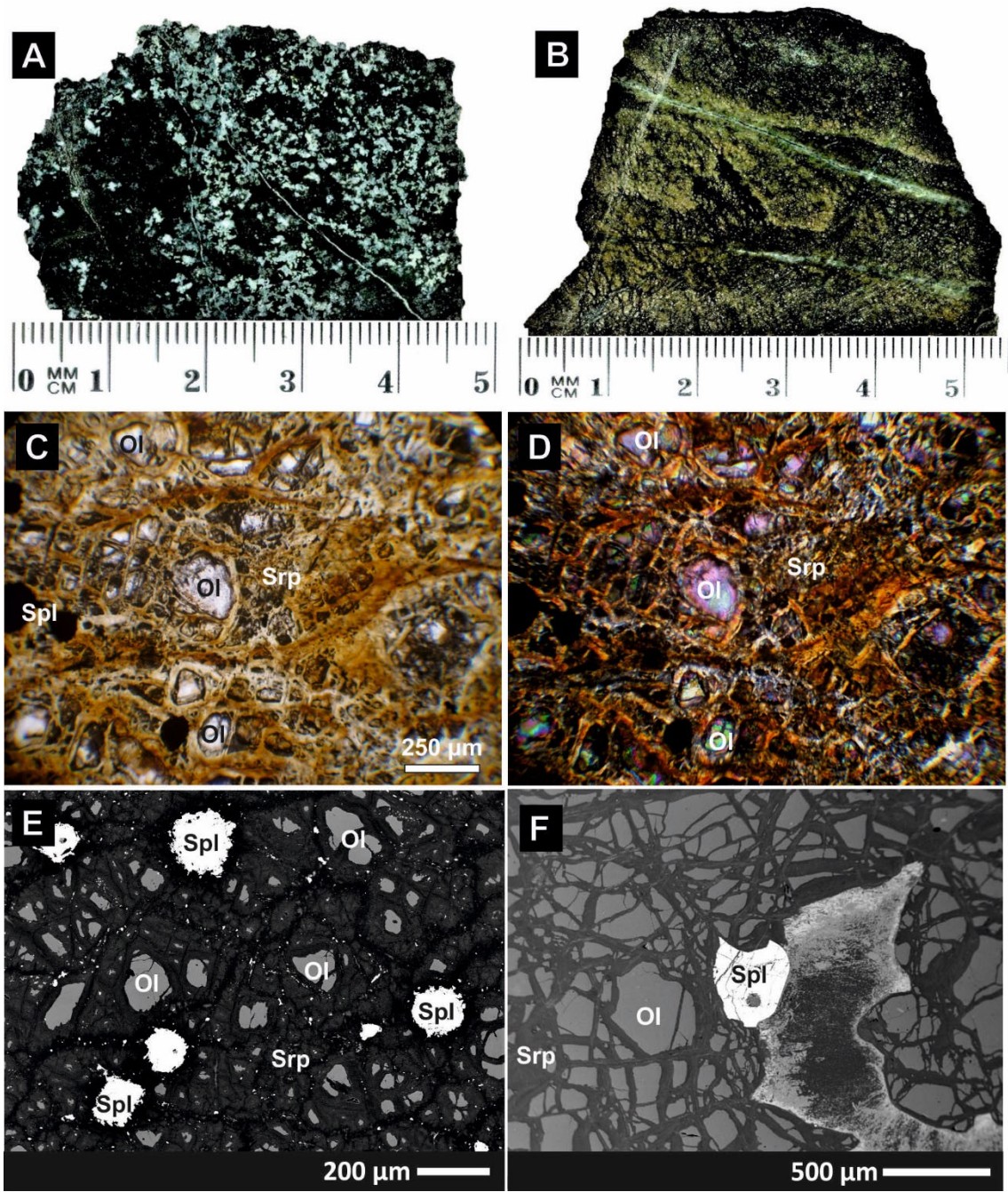

**Figure 8.** Macroscopic (A,B), PL microscopic (C,D), and BSE (E,F) images of dunite samples: (**A**) Feldspatization of dunite to troctolite P19A2 at the contact with Cpx–Pl gabbro dyke P19A1. (**B–E**) Dunite P11A. (**F**) Dunite P18D. Picture C at *II* P; D at *X* P.

### 4.2. Petrography of Plagioclase Peridotites and Troctolites

Here, we describe the main macro- and microscopic features of the rocks that may have been formed due to the percolation of inferred fractionated Ca-Si-Al-enriched melts through lherzolites, harzburgites, and dunites, forming Pl peridotites and troctolites. The troctolite layer in the Paklenica Valley (P18A–C,E) strictly follows a dunite layer (P18D) within peridotite (P29A2).

### 4.2.1. Plagioclase peridotites

Part of the lherzolites at Gostilj peak (P8A) contain Pl. The presence of Pl in peridotites was detected from PL microscopy and EPMA BSE images. These images are characterized by the pervasive distribution of Pl in the peridotite structure. Plagioclase is partly replaced by Ab, H-Grs, and Czo (Figure 9A,B).

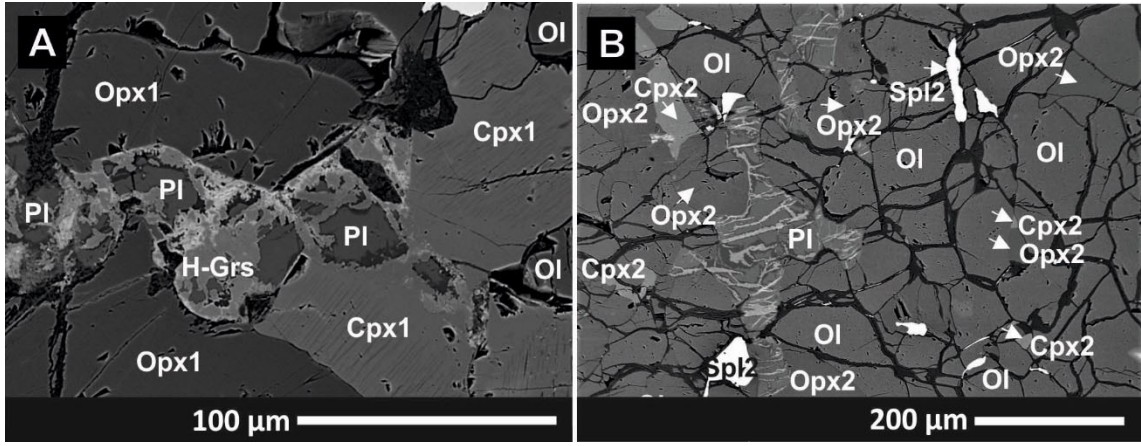

**Figure 9.** Plagioclase peridotite (s. P8A, lherzolite). (**A,B**) Relics of An-rich Pl in Ab-Czo-H-Grs aggregate.

### 4.2.2. Troctolite in Dunite Layers

The dunite layer in Donja Paklenica Valley peridotite contains banded troctolite crosscut by a dolerite dyke (Figure 10A,B). The transition from original Ol-Spl dunite to Ol-Spl-Pl-Cpx troctolite is quite easily observable (Figure 10C–E). Some troctolite parts are almost homogeneous, but Ol-rich (black) pods after the dunite are still visible (Figure 10E). The other ones show a banded structure due to the alternation of dark Ol-rich and pale Pl-rich bands. The youngest appear to be aplitic to pegmatitic Pl-rich veins, which even penetrate the dolerite dyke (Figure 10A,B,F).

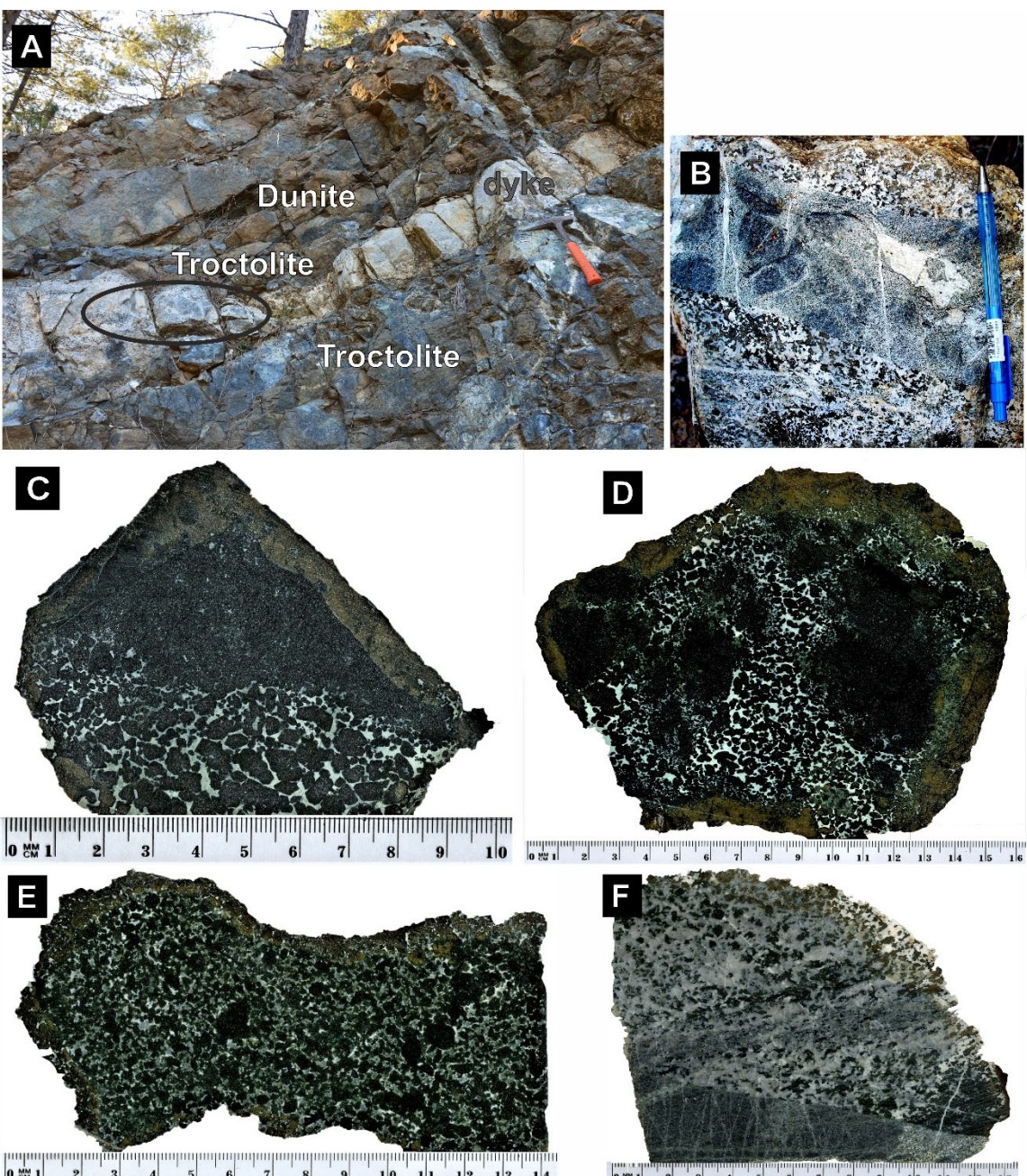

**Figure 10.** An outcrop in Donja Paklenica Valley peridotite with a dyke at the dunite-troctolite boundary. (**A**) Troctolite is crosscut by dolerite dyke (a detail view on the circle domain in B). (**B**) Dolerite dyke (P18G2) crosscutting troctolite (from A). (**C–E**) Secondary Pl enrichment of Ol-Spl dunite (black parts) and the formation of Ol-Pl-(±Cpx) troctolite (the area above the dyke in A). (**F**) Troctolite crosscut by fine-grained Pl-rich vein (at the bottom).

The microstructure of troctolite P18A documents the irregular distribution of Pl in the Ol-Spl matrix (Figure 11A–D). The contact between Ol and Pl is systematically accompanied by newly formed interstitial Cpx enclosing Ol and/or Spl (Figure 11A–D). Spinel and Ol show partial dissolution in contact with Pl (Figure 11C,D), and Cpx overgrows Spl in coronas (Figure 11E). The grain boundaries are accompanied by hydrated phases such as Phl, Amp (Prg, Krs, Fkrs), Chl, Czo, and Cpx, which extend from the Spl rim into the Spl interior (Figure 11E–H). Plagioclase underwent albitization and further alterations including the formation of Str aggregates in Ab (Figure 11G,H). Olivine is serpentinized (Figure 11A–E,G).

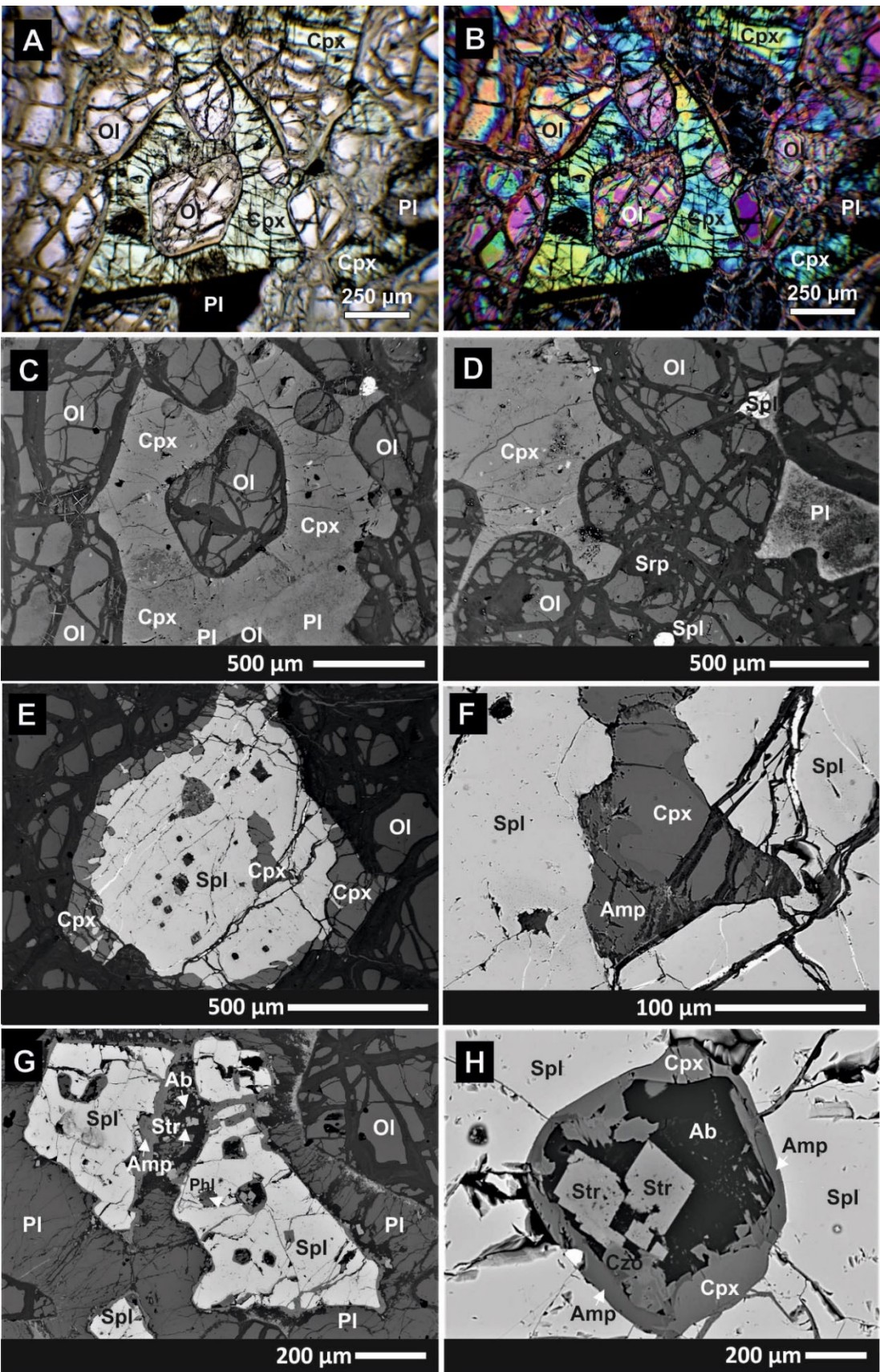

**Figure 11.** PL microscopic (A,B) and BSE (C–H) images from troctolite (P18A). (**A–D**) Newly formed Pl and Cpx enclosing Ol and Spl. (**E–H**) Partial replacement of Spl by Cpx (outer and inner coronas), Amp (Krs), Phl, Czo, Ab, and Str in Ab. Picture A at *II* P; B at *X* P.

*4.3. Petrography of Gabbros, Plagiogranites, and Basalts*

4.3.1. Gabbros and Plagiogranites

A gabbro body (samples P10A–C) crosscut by a basalt dyke (P10D) was identified from the ca. 100 m borehole core profile (Figure 12A,B). Gabbro is characterized by an alternation of Cpx-Pl, Cpx-Pl-Amp, and Pl-rich types. These gabbros have an ophitic texture and contain primary magmatic porphyric Cpx, Pl, and green Amp1(Mhb, Prg). Pyroxene and Amp1 margins are partly replaced by bluish Amp5(Tr, Act) and Chl (pale green) aggregates. Plagioclase is partly replaced by Ab and Czo, and weakly altered. Epidote-Amp5-Chl aggregates occur in grain boundaries of magmatic phases (Figure 12C,D). Similar litho-types also occur in ophiolitic breccia overlying peridotites SE of Maglaj at Bočinja village (Figure 2). One gabbro fragment (P13B1) from the breccia is crosscut by a leucogabbro vein (P13B2; Figure 12E). A larger plagiogranite body (P5; Figure 12F) occurs in the northern part of the OOC (Figure 2), in contact with peridotites (P4, P6).

Here, and in the next chapter, we use Amp indexes (1–5), which are defined in chapter 4.3.3., where some of the dolerite dykes (P17C) contain 5 Amp generations, and all of them are classified in chapter 4.5.5.

The BSE image of gabbro (P13B1) documents secondary Amp5 aggregates in Chl, the albitization of Pl, and local replacement of Ilm by a $TiO_2$ phase and Ttn (Figure 13A).

Plagiogranite (P5B) is composed of Pl, Qz, Amp, Bt, and Ilm. Deformed Bt flakes coexist with needle-like Amp1(Wnc) aggregates. Biotite is partly replaced by Chl and Ttn, and Ilm is replaced by $TiO_2$ and Ttn. Plagioclase is Ab (Figure 13B).

The leucogabbro vein (P13B2) in gabbro (P13B1) consists of Cpx and Amp1(Mhb, Prg) in a Pl-dominated matrix. Secondary Amp5(Tr, Act) is associated with Chl, Ab, Czo, Ttn, and a $TiO_2$ phase (Figure 13C,D).

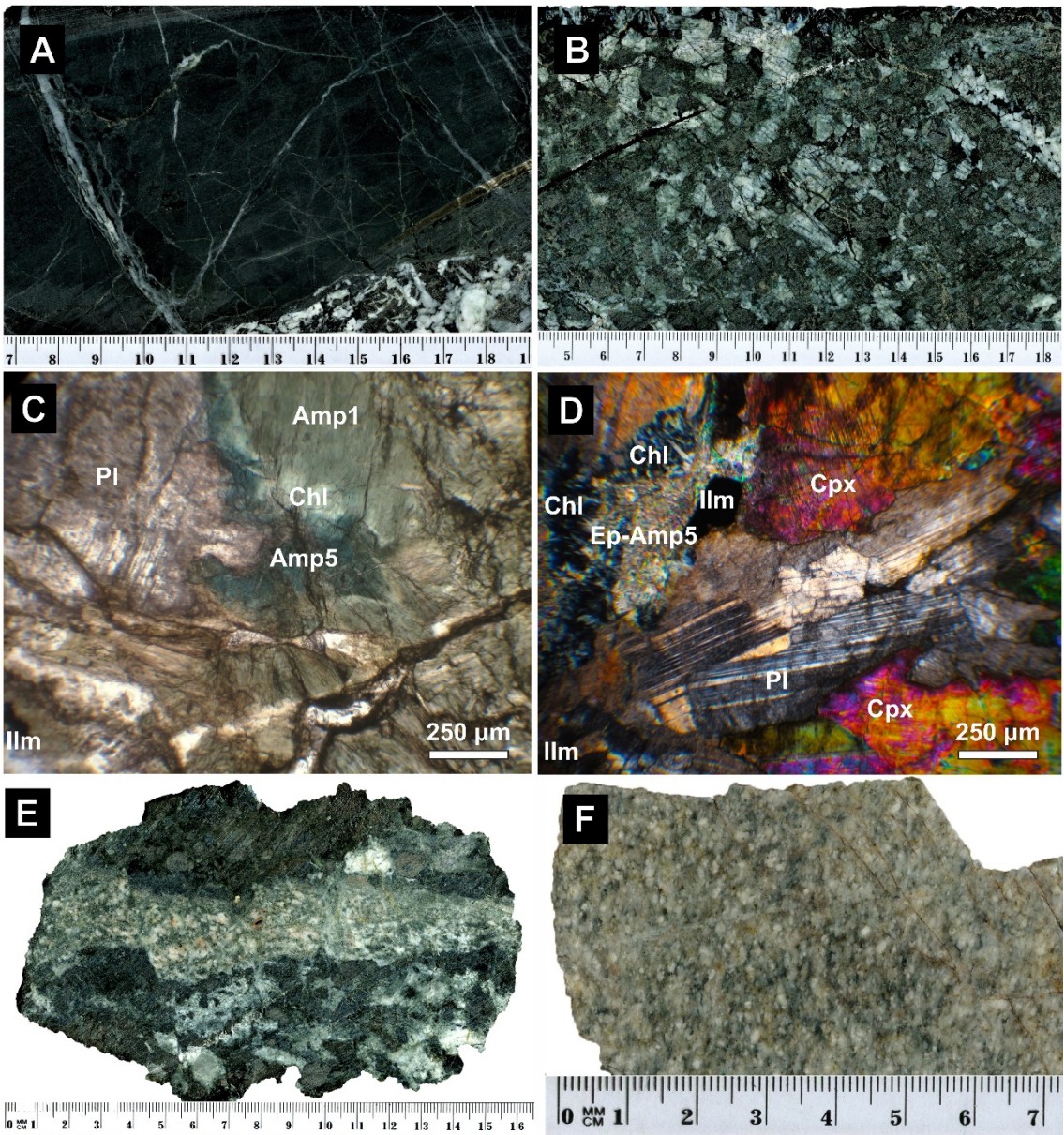

**Figure 12.** Gabbros and a plagiogranite from inferred gabbro layer of the OOC oceanic crust from a borehole core (P10A–C), ophiolitic breccia (P13B1, P13B2), and an intrusive body (P5). (**A**) Gabbro (P10A) crosscut by a basalt (dark) dyke (P10D). (**B**) Cpx-Pl-Amp gabbro (P10C). (**C**) PL microscopic image of Cpx-Pl-Amp-Ilm gabbro (P10C) with blue-green rim of Amp5(Tr, Act) on Amp1(Mhb, Prg). (**D**) PL microscopic image of Cpx-Pl gabbro (P10B) with secondary fine-grained aggregate of Amp5, Czo, Ep, Chl. (**E**) Gabbro (P13B1) crosscut by a leucogabbro vein (P13B2). (**F**) Plagiogranite (P5). Picture C at *II* P; D at *X* P.

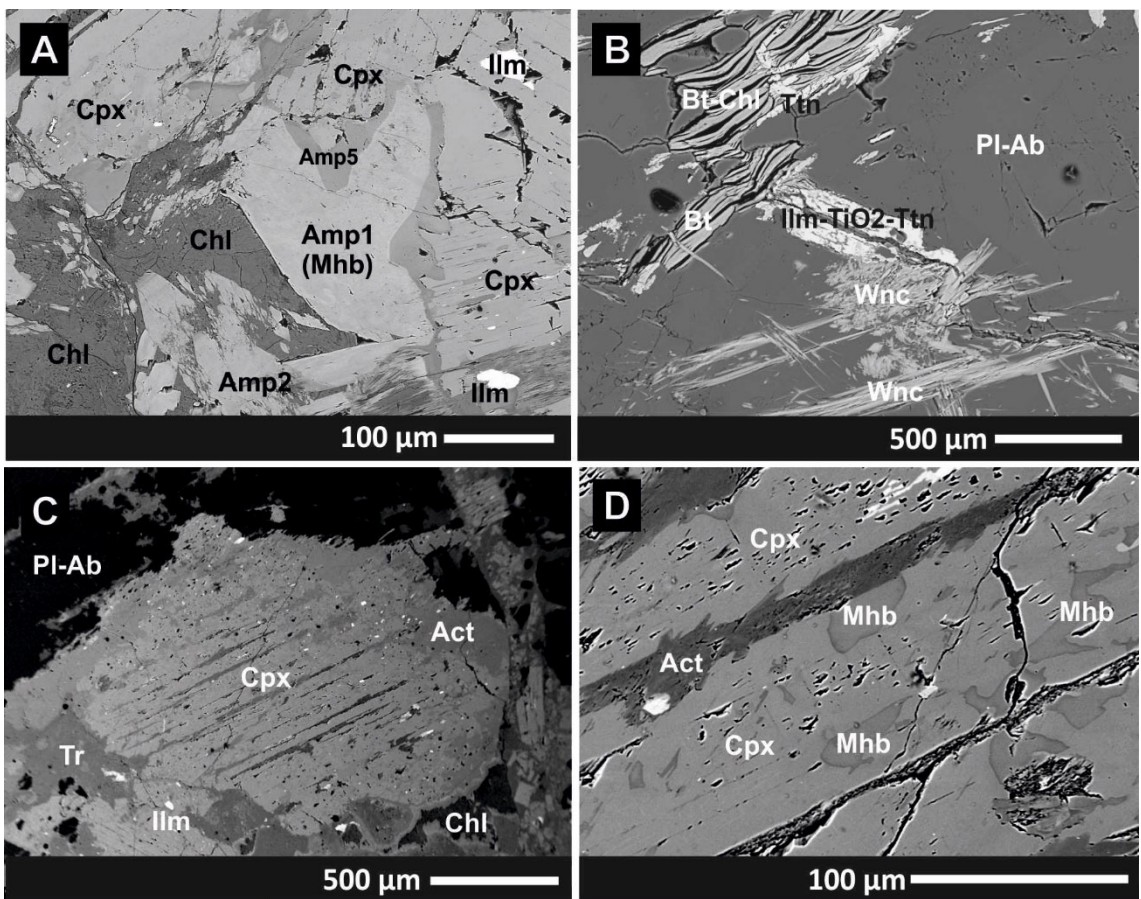

**Figure 13.** BSE images of gabbros and a plagiogranite from inferred gabbro layer of the OOC oceanic crust. (**A**) Cpx-Pl-Amp1(Mhb)-Ilm gabbro (P13B1) with newly formed aggregate of Amp5, Chl, Ab, Ttn. (**B**) Plagiogranite (P5B) with deformed chloritized Bt associated with needle-like Amp1(Wnc) aggregates in the secondary assemblage of Ttn, TiO₂, and Chl. (**C**) Leucogabbro (P13B2) composed of Cpx, Pl, Amp1 (visible in D), and Ilm. (**D**) Leucogabbro (P13B2) Cpx partly replaced by Amp1(Mhb, Prg) and Amp5(Act).

### 4.3.2. Gabbro Dykes

Here, we describe the main macro- and microscopic features of the gabbro dykes through peridotites. These rocks mostly have a discordant position to peridotite planar structures, such as, for example, the dykes in the Paklenica Valley (P29A1,C1) on the western side of the OOC. Exceptionally, they intruded dunitic layers in peridotite (P19A1, in the north).

We found numerous crosscutting gabbroic dykes in peridotite, from micro-gabbros (an equivalent of dolerites in grain size) to gabbro-pegmatites (Figure 14A–F). The dyke minerals are randomly oriented in the dyke centre, while the dyke margins have weakly oriented flow texture. Coarse-grained, pegmatitic types display a grain-size variability, from coarse grained centimeter-sized crystals in the dyke core to medium/finer-grained millimeter-sized crystals at the margins (Figure 14C,D). Most of the dykes have well-preserved magmatic structure (Figure 14D,E), whereas mylonitic banding was only locally observable (Figure 14F).

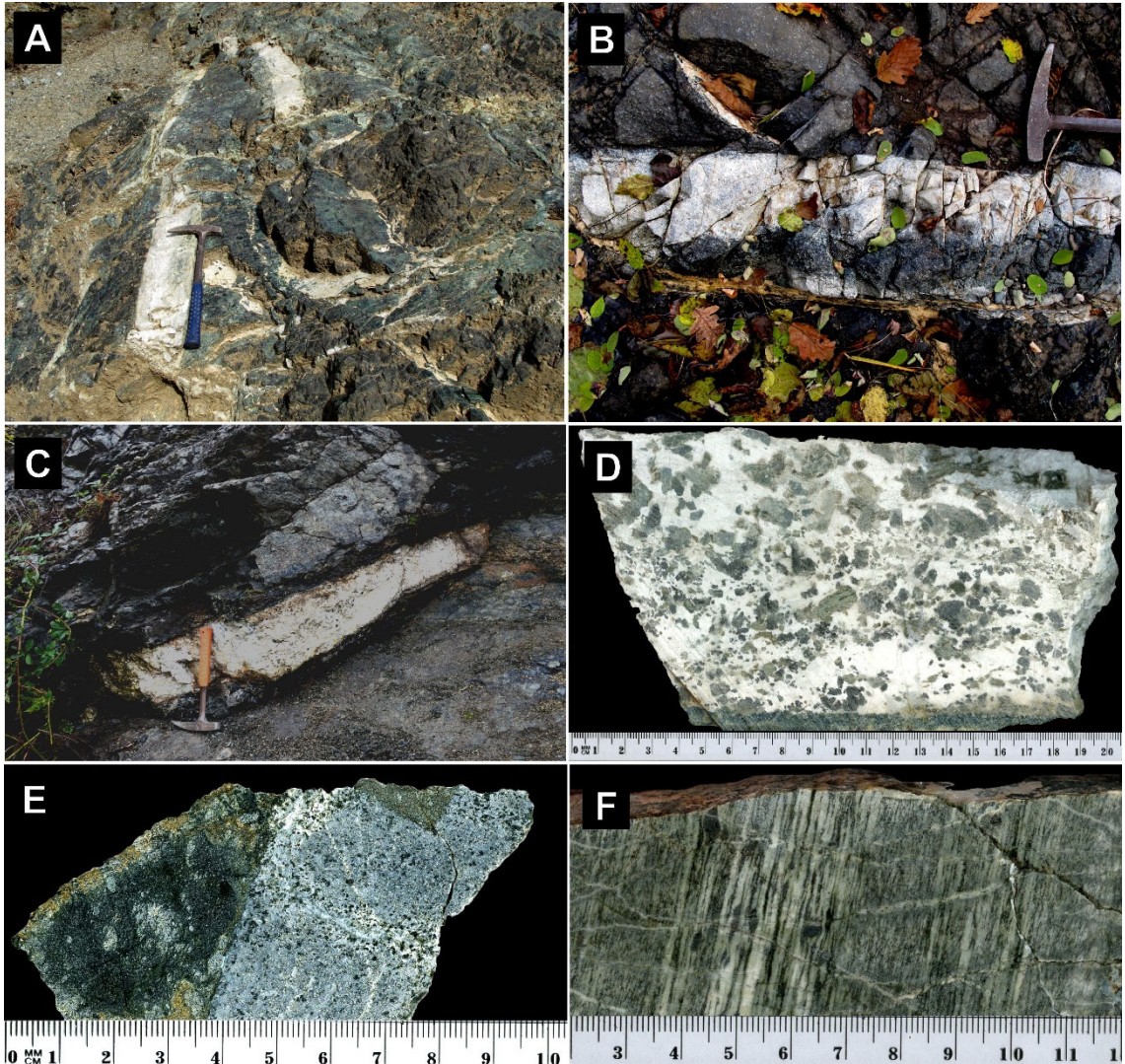

**Figure 14.** Gabbro dykes in peridotite. (**A**) Mylonitized Cpx-Pl gabbro dyke (P23A) in host serpentinized harzburgite. (**B**) Plagioclase-rich gabbro dyke (pale, P19A1) intruded a dunitic layer (dark, P19A2) within the host harzburgite. (**C**) Boudinaged gabbro-pegmatite (P29C1) dyke in harzburgite. (**D**) Cut surface of gabbro-pegmatite (P29C1) shows Cpx and Pl grain-size reduction towards interface with the harzburgite host (P29C2) at the bottom. (**E**) Cut surface of micro-gabbro dyke (P29A1) in harzburgite (P29A2) with a cm-size xenolith of the host peridotite (on the top). (**F**) Cut surface of Cpx-Pl gabbro mylonite (P23A) with rotated σ-type Cpx porphyroclasts in the mylonitic banded structure.

Gabbroic dykes are typically undeformed (Figure 15A,B), and most of them have well-preserved ophitic microstructures composed of Cpx and Pl, accessory Ilm and Ap, and rare Zrn. These dykes usually do not contain magmatic Amp. The secondary phases such as Amp5, Ab, Ep, Czo, and Chl follow the grain boundaries of magmatic minerals, and Ilm is replaced by $TiO_2$ and Ttn. The mylonitized dyke P23A shows typical core–mantle structures of Cpx porphyroclasts (with subsolidus exsolution lamellae) wrapped by the dynamically recrystallized fine-grained Cpx aggregates. Clinopyroxene is chloritized, and Pl bands are altered to H-Grs (Figure 15C).

Dyke rims often show a mechanical interaction or assimilation features along the contact with dunite layers in harzburgite (Figures 8A,14B, and 15D), or directly with a host harzburgite (Figure 15E,F).

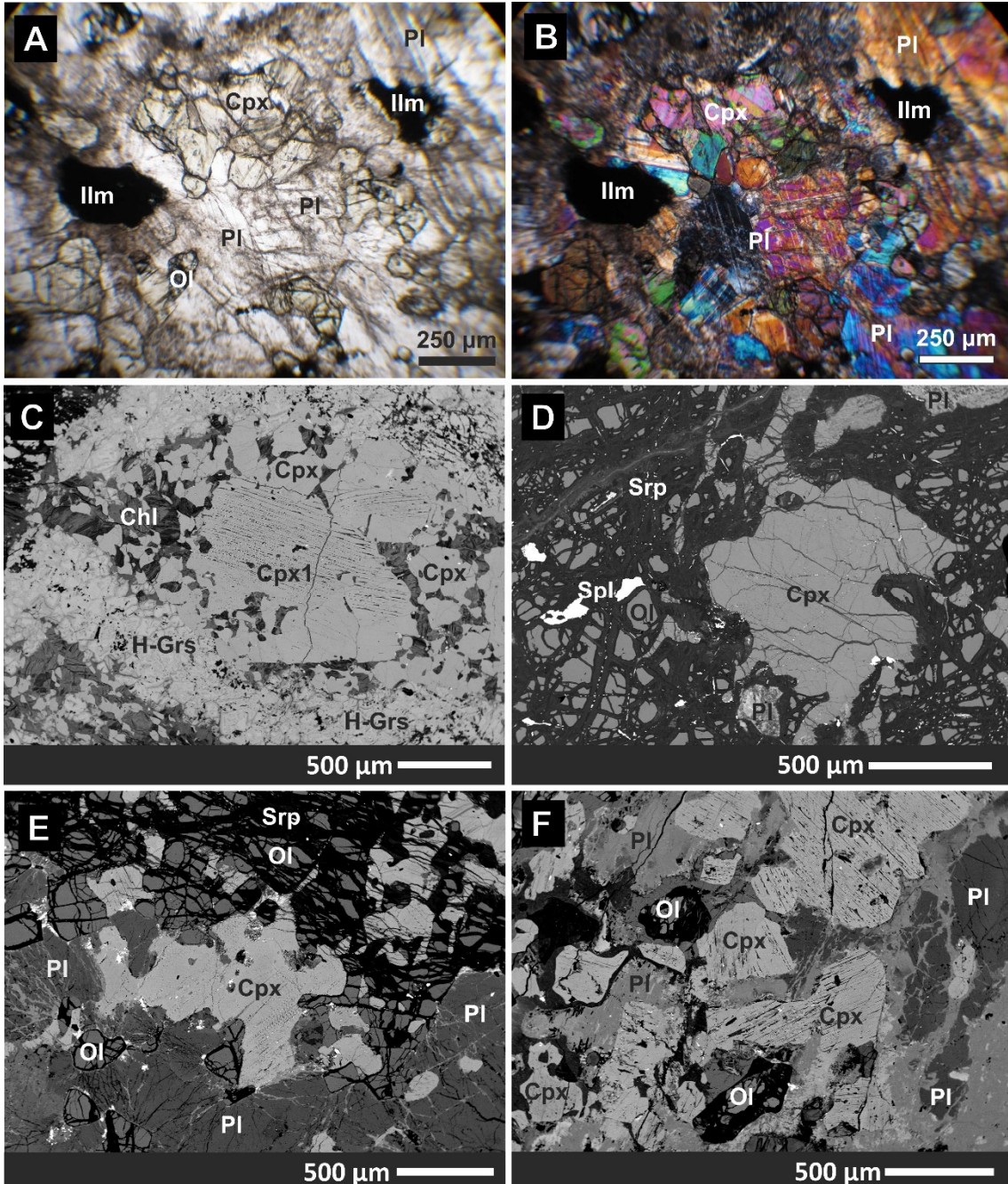

**Figure 15.** PL microscopic (A,B) and BSE (C–F) images of gabbro dykes. (**A,B**) Cpx–Pl–Ilm gabbro (P29A1). (**C**) Gabbro-mylonite (P23A) with signatures of dynamic recrystallization of Cpx. Alteration of a Pl band to H-Grs. (**D**) Cpx-Pl percolation through the Ol-Spl matrix of a dunite layer in peridotite. (**E**) Contact and partial mixing of a host harzburgite (P29A2) with the Cpx-Pl-Ilm gabbro dyke (P29A1). (**F**) The central part of the Cpx-Pl-Ilm dyke (P29A1) contains rare, altered Ol xenocrysts detached from the host peridotite (P29A2). Picture A at *II* P; B at *X* P.

### 4.3.3. Doleritic Gabbros, Dykes, and Basalts

Dolerites are finer-grained (Figure 16A,B) in comparison with medium- to coarse-grained gabbros. They occur in the form of massive intrusions in serpentinized peridotites (P15, P16) or in the form of crosscutting dykes through peridotites (P9, P10, P17A-C) and rarely troctolites (P18G2). The dolerites are usually composed of Cpx, Amp1, and Pl, with accessory Ilm, less Ap, and Ttn. Magmatic pyroxenes and amphiboles are chloritized, and Pl is albitized or replaced by Ep-Czo. Medium- to fine-grained dolerites (P17A–C) in the middle part of the Paklenica Valley (Figure 16A) occur as dykes or lens-shape bodies

within serpentinized peridotites. Similar rock types also occur in an ophiolitic breccia SE of Maglaj in the form of metric fragments (P13C, P13D).

So far, we discovered only one dolerite dyke containing Ol and Opx besides Cpx, brown Amp1, Pl, Ilm, and Ap in well-preserved magmatic texture (P9, from the northeastern part of the OOC; Figure 16B–D).

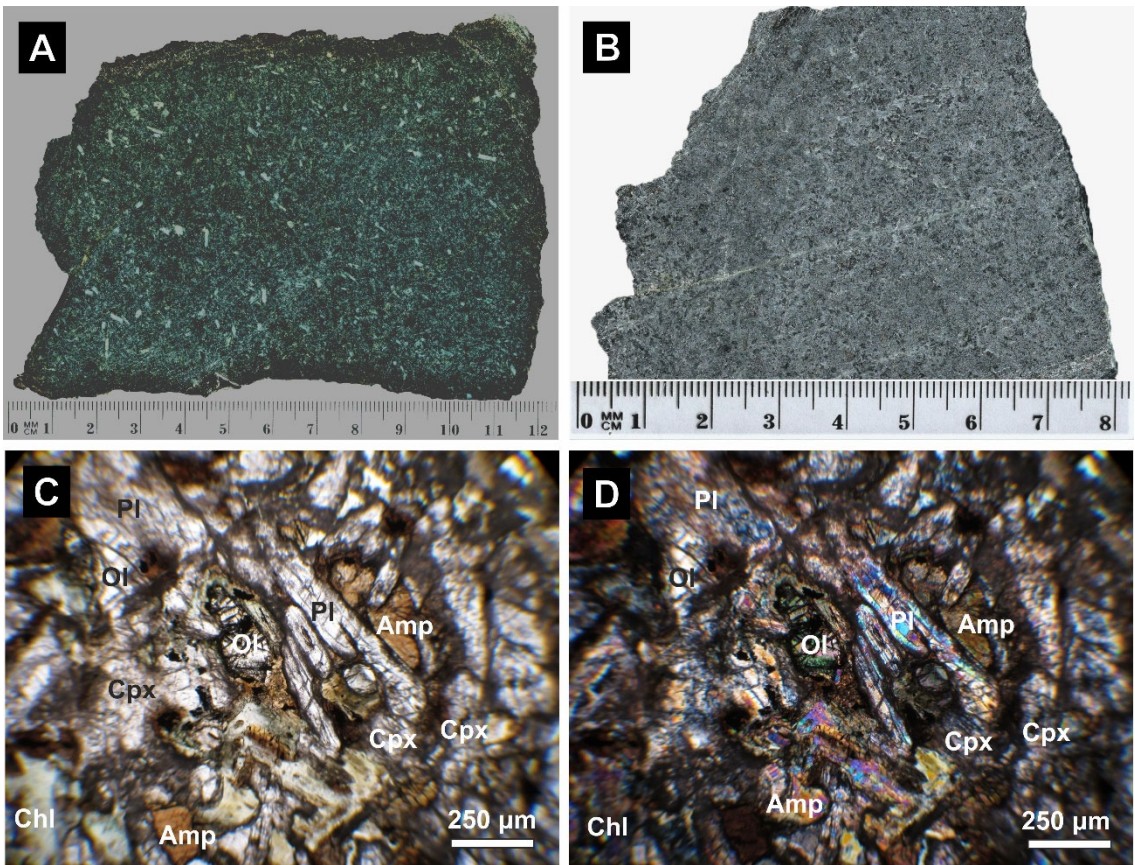

**Figure 16.** Macroscopic (A,B) and PL microscopic (C,D) images of dolerites. (**A**) Cpx-Pl-Amp dolerite (P17C). (**B–D**) Ol-Opx-Cpx-Amp–Pl dolerite (P9). Picture C at *II* P; D at *X* P.

The BSE image of Ol-dolerite dyke (P9) crosscutting peridotite documents Ilm exsolutions in porphyric Cpx. Accessory Ilm is partly replaced by TiO₂ and Ttn, Pl is albitized at the rims, Cpx and Amp1 (Prg, Hst) are partly replaced by Amp5 (Tr, Act) and Chl, and Ol is serpentinized (Figure 17A).

The BSE image of the Cpx-Pl-Amp1-Ilm dolerite dyke (P17C) in peridotite displays magmatic texture defined by porphyric Cpx, green Amp1(Mhb), and Pl, with accessory Ilm. There are four types of younger Amp (2–5) in this dyke: (1) Needle-like aggregates of late-magmatic Amp2 (Mhb) occur at the rims of Amp1 and in grain boundaries of Cpx, Amp1, and Pl; transitions from Amp1 to Amp2 of the same chemical composition are observable; (2) dark core hypidiomorphic to idiomorphic Amp3 (Wnc, Ktp), which is overgrown by (3) a pale Amp4 (Gru) rim, and (4) the last Amp5 (Tr, Act) generation occurs with Chl, Czo, and Ab (Figure 17B).

The BSE image of Cpx-Pl-Amp-Ilm dolerite (P13C) from ophiolitic breccia documents late-magmatic Amp2 (Mhb) needle-like aggregates in grain boundaries of Cpx, Amp1, and Pl (Figure 17C).

The BSE image of the Amp-Pl-Ilm basalt dyke (P10D) from a borehole core (Figure 17D) shows a well-preserved magmatic texture almost without the secondary phases. It resembles a basalt fragment from ophiolitic breccia (P13F) composed of Cpx, Pl, Ilm, and Ttn in the ophitic texture.

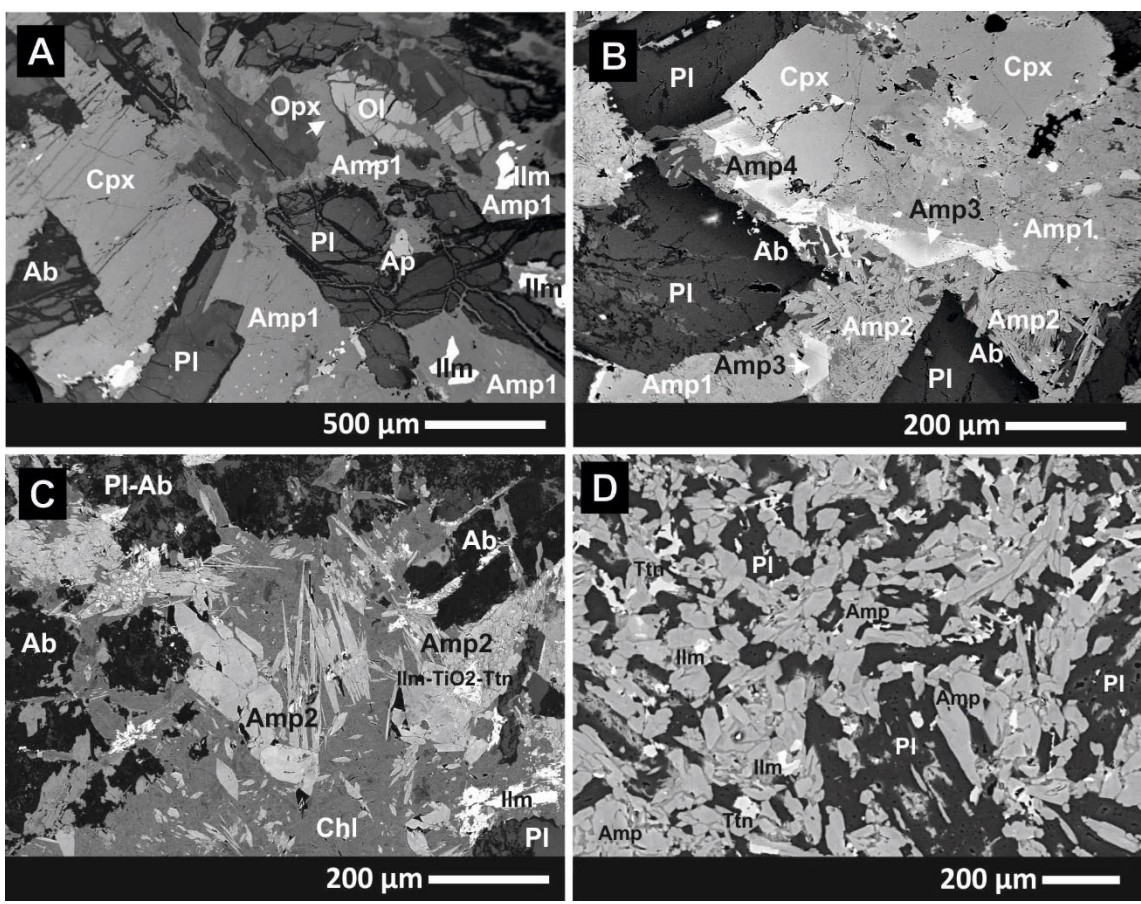

**Figure 17.** BSE images of dolerites (A–C) and a basalt (D). (**A**) Ol-Opx-Cpx-Pl-Amp dolerite (P9). (**B**) Cpx-Pl-Amp-Ilm dolerite (P17C) with the primary Amp1 and the secondary Amp2–4 types. (**C**) Cpx-Pl-Amp-Ilm dolerite (P13C) with the secondary Amp2, Chl, Ab, TiO2 a Ttn. (**D**) Amp-Pl-Ilm basalt (P10D).

*4.4. Petrography of Amphibolites*

Amphibolite samples P12A, P12B, and P10F from the inferred hanging walls of ophiolitic thrust sheets were found from a surface outcrop (P12A) and the two borehole cores (P12B, P10F). Their texture changes from a coarse-grained, slightly oriented granoblastic to fine-grained schistose (Figure 18A,B). The coarse-grained part is composed of metamorphic Amp (Mhb, Prg) and Pl (Figure 18C,D) and the relics of magmatic Cpx in idiomorphic Czo porphyroclastic pseudomorphs (Figure 18E). The fine-grained shear bands are composed of porphyroclasts of Mhb or Prg and albitized Pl. The hornblendes are partly replaced by the Tr and Act aggregates associated with Chl, Phg, Ab, Zoi, Ep, Czo, and Ttn (Figure 18F).

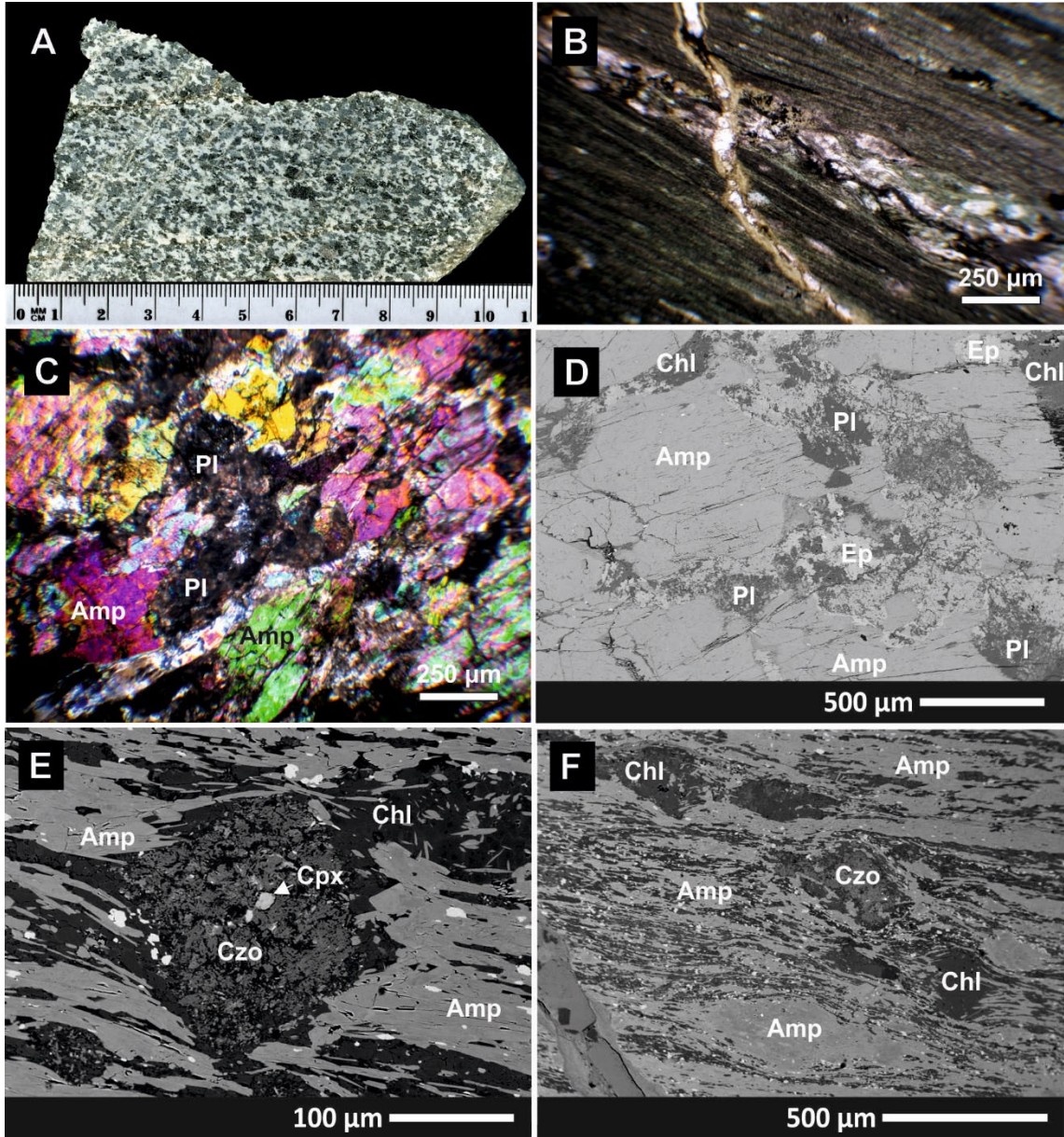

**Figure 18.** Macroscopic (A), PL microscopic (B,C), and BSE (D–F) images of amphibolites from the inferred ophiolitic thrust-sheet hanging walls. (**A**) Slightly anisotropic amphibolite texture in coarse-grained amphibolite part composed of Amp and Pl with magmatic Cpx relics (P12A). (**B**) Fine-grained shear bands of amphibolites (P10F). (**C,D**) Texture of coarse-grained amphibolite layers with Amp-preferred orientation (P12A). Recrystallization aggregates of Chl, Tr, Act, Ep, Czo, Ab, and Ttn in grain boundaries of Mhb, Prg, and Pl. (**E**) Porphyroclast of idiomorphic Czo pseudo-morph with the inferred magmatic Cpx relics in the middle (P10F). (**F**) Porphyroclasts of Mhb and Prg in sheared matrix of recrystallized Mhb and Prg, associated with Tr, Act, Ep, Czo, Phg, Ab, and Chl (P10F). Picture B at *II* P; C at *X* P.

### 4.5. Mineral Chemistry

4.5.1. Olivine in Peridotites, Troctolites, and a Dolerite Dyke

Ni (*apfu*) content in Ol generally decreases from 0.009 to 0.003 with a Mg# decrease from 92 to 83 in ultramafics and troctolites. Most peridotites contain Ol with Mg# in a narrow interval between 90 and 92, and Ni ranging from 0.006 to 0.009 *apfu*. The lowest Ni content of 0.001 *apfu* was found in Ol from dolerite dyke P9, which also has the lowest Mg# of 65 (Figure 19). Figure 19, on the other hand, documents an increase of Fe in Ol from troctolite (P19A2) and, similarly, in Ol xenocrysts from gabbro dykes (P29A1, P29C1)

along interfaces with hosting harzburgites (P29A2, P29C2). The characteristic Ol FeO content in peridotites is 8–10 wt.%, whereas a higher Ol content up to 16.5 wt.% was found in troctolites and peridotites in interfaces with gabbroic dykes (Tables 2 and 3). Olivine in dolerite dyke P9 has the highest FeO content of 31 wt.%.

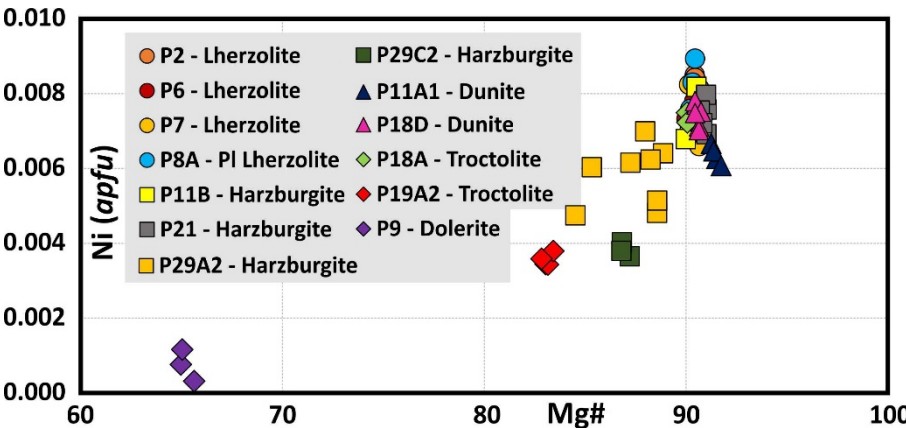

**Figure 19.** Relationship between Ni (*apfu*) and Mg# = [Mg/(Mg + Fe$^{2+}$) × 100] atomic ratio in olivine.

### 4.5.2. Spinel (1–3) in Peridotites, Troctolites, and a Dolerite Dyke

Magmatic Spl group has a broader chemical composition of spinel, Mg-chromite, chromite, and hercynite. The Spl1 Cr# is in the range of 45–56, and Mg# from 32 to 70 in ultramafics and troctolites (Figure 20A; Tables 2 and 3). While the Spl Cr# of dunites and troctolites are comparable, the Spl TiO$_2$ and Al$_2$O$_3$ are higher in troctolites. Dolerite dyke P9 has much lower Mg#, but much higher TiO$_2$ compared to ultramafics and troctolites.

There is a distinct decrease in Cr# and TiO$_2$, while Al$_2$O$_3$ increases, from Spl1 to Spl2 and 3 in ultramafics (Figure 20A,B; Tables 2 and 3). The interesting aspect is the Ni content increase from Spl1 to Spl2 and 3, which may indicate Ol dissolution.

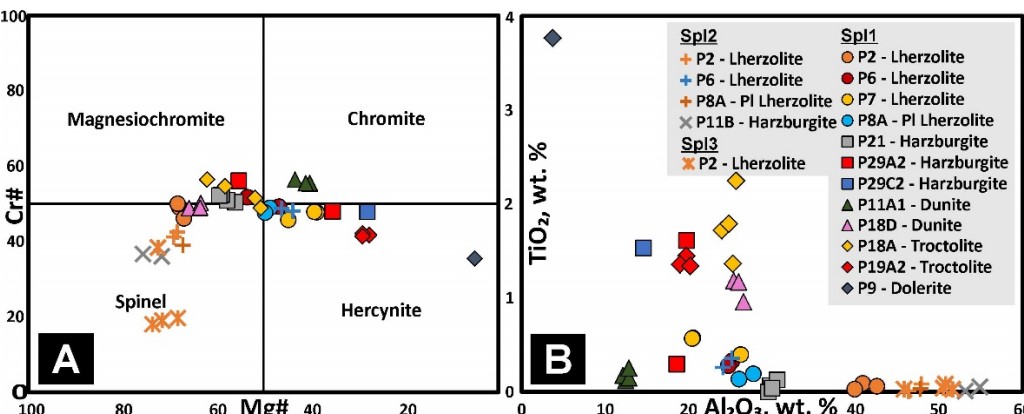

**Figure 20.** Spinel group chemistry. (**A**) Relationship between Mg# = [Mg/(Mg + Fe$^{2+}$) × 100] atomic ratio and Cr# = [Cr/(Cr + Al) × 100] atomic ratio. (**B**) Relationship between Al$_2$O$_3$ and TiO$_2$ spinel content (in wt.%). The individual spinel-group end-member mineral terminology is from [54].

### 4.5.3. Orthopyroxene (Opx1–2) in Peridotites and a Dolerite Dyke

Magmatic orthopyroxene is enstatite (Figure 21A). Ultramafics have high-Al Opx1 between 2 and 6 wt.% Al$_2$O$_3$, whereas Opx2 contains only approximately 2 wt.% Al$_2$O$_3$. Dolerite dyke P9 has approximately 1 wt.% Al$_2$O$_3$. Ultramafics have Opx with a high Mg# of 88–92, whereas the dolerite dyke has much lower Mg# of 72–74 (Figure 21B).

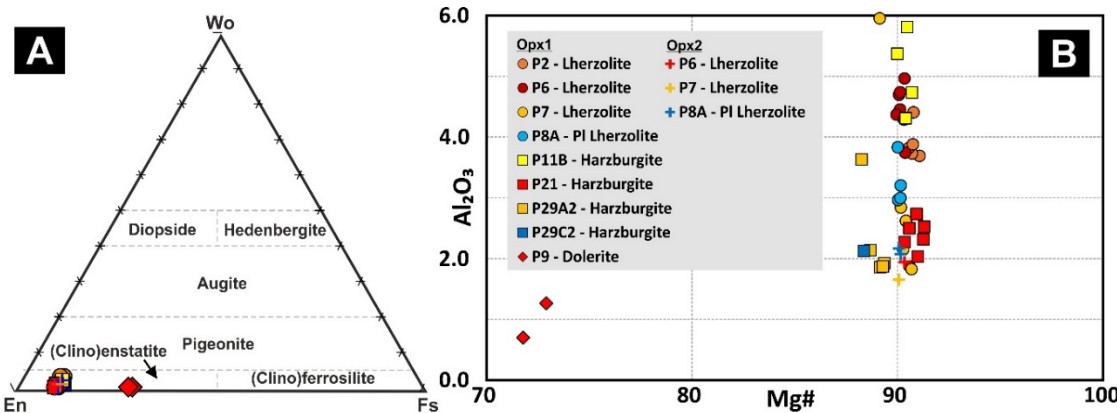

**Figure 21.** Orthopyroxene chemistry. (**A**) Opx Classification from [55]. (**B**) Relationship between $Al_2O_3$ and Mg# = [Mg/(Mg + Fe$^{2+}$) × 100] atomic ratio in Opx.

4.5.4. Clinopyroxene (Cpx1–3) in Peridotites, Troctolites, Gabbros, Dolerites, Basalts, and Amphibolites

Magmatic clinopyroxene (Cpx1–3) of ultramafics is diopside (Figure 22A), whereas mafics contain diopside to augite (Figure 22B). Ultramafics have high-Al Cpx1 between 2 and 6 wt.% $Al_2O_3$, whereas Cpx2 and Cpx3 contain mostly 2–3 wt.% $Al_2O_3$. Mg# varies between 90 and 100 (Figure 22C).

Mafics have a more variable content of $Al_2O_3$, ranging from 0.5–6.5 wt.%, and similarly, Mg# of 45–100 (Figure 22D).

Amphibolites (P10F) from an inferred ophiolitic thrust-sheet hanging wall contain relic magmatic Cpx, hedenbergite (Figure 22B).

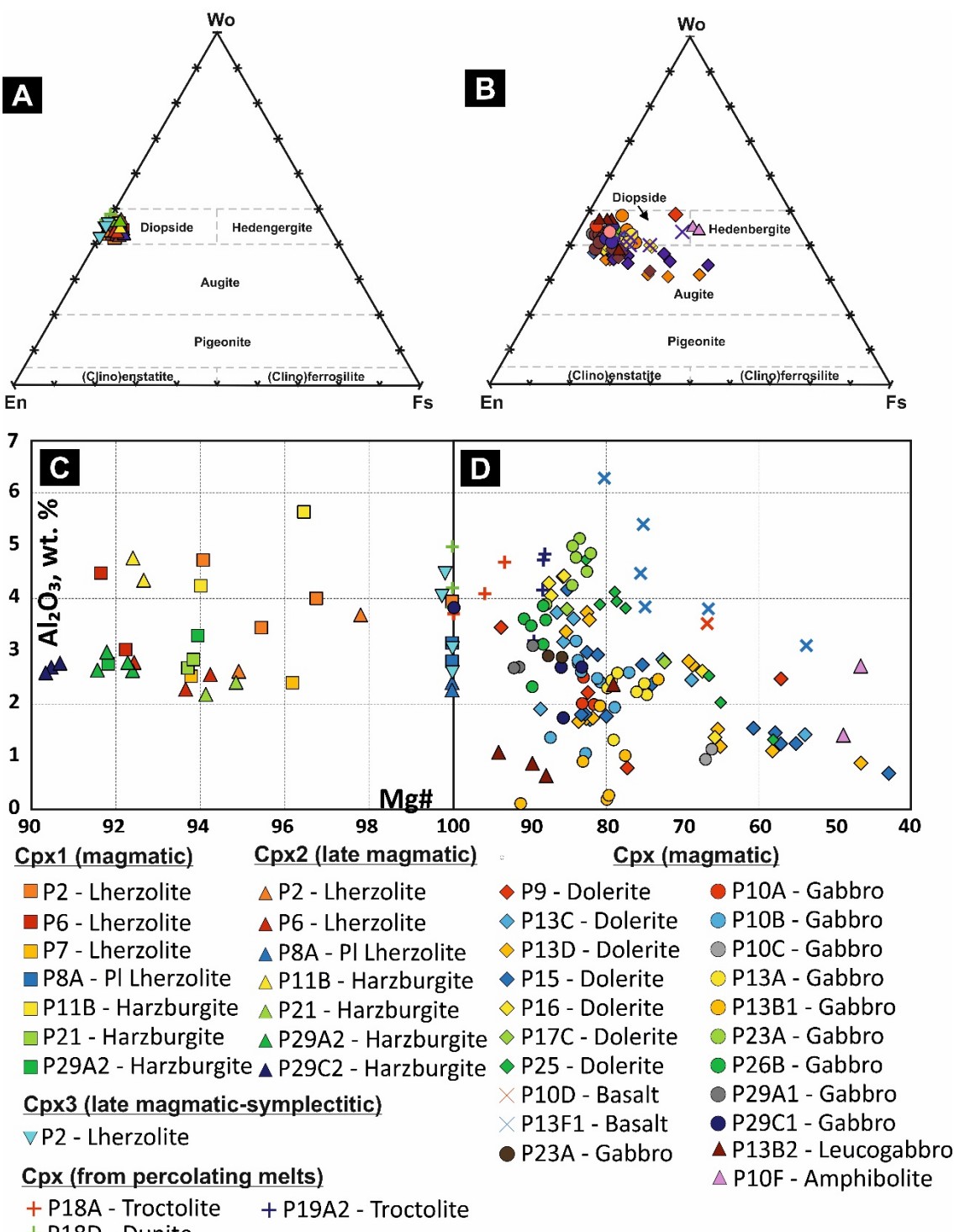

**Figure 22.** Clinopyroxene classification after [55]. (**A**) Ultramafic rocks. (**B**) Mafic rocks. Relationship between $Al_2O_3$ and Mg# = [Mg/(Mg + $Fe^{2+}$) × 100] atomic ratio in Cpx for (**C**) ultramafic rocks and (**D**) mafic rocks.

### 4.5.5. Amphibole (1–6) in Ultramafics, Mafics, and Amphibolites

A few successive Amp types with different chemical compositions were identified in the rock textures using EPMA (Figure 23; Tables 2 and 3). Ultramafics typically do not contain Amp. A secondary Amp-Mhb was found only in P21 harzburgite, whereas primary and secondary Amp are common in gabbros, dolerites, and plagiogranites. Troctolites (P18A) and dunites (P18D) may contain secondary Amp too.

*Magmatic Amp1*

Clinopyroxene-Pl gabbros (P10B, P13A, P29A1, P29C1) and dolerites (P15, P25, P26B) do not contain coarse-grained Amp1, only prismatic to needle-like Amp2 (Mhb) aggregates or late generation of Amp5 (Tr, Act) in Chl (see the next paragraph). Magmatic Amp1 was determined in almost all gabbroic rocks. Gabbro samples from borehole cores contain Amp1-Mhb to Prg (P10A, P10C). Crosscutting basaltic dyke P10D contains Amp1 (Mhb, Prg, and Hst). Furthermore, some other representative samples of gabbros (P13B1) and dolerites (P13C, P13D, P16, P17B, P17C) contain Amp1-Mhb as the principal Amp type. However, brown Amp1 in P9 dolerite dyke is Prg or Hst. Tonalitic plagiogranite P5B contains Wnc and less Prg, whereas Cpx-bearing leucogabbro P13B2 has Mhb to Prg (Figure 23; Table 3).

*Late magmatic Amp2*

The most common are prismatic to needle-like Amp2-Mhb (Figure 23; Tables 2 and 3) aggregates in gabbros and dolerites. This Amp generation formed after crystallization of Amp1. It occurs in grain boundaries of magmatic Amp1, Cpx, and Pl in gabbros (P10A, P10C, P13B1) and dolerites (P13C, P15, P16, P17B, P17C). There are commonly observable gradual transitions from Amp1 grain to Amp2 aggregates, both having the same chemical composition of Mhb (e.g., P13B1, P13C, P17C; Figures 13A, 17B,C, and 23). Amphibole2 spots may cover Cpx or grow along a distinct Cpx crystallographic plane system (P10B).

*Secondary Amp3, 4 (from inferred percolating melts/fluids)*

In general, this Amp generation is rare. The best example is the dolerite dyke P17C. This dolerite contains hypidiomorphic to idiomorphic porphyric Amp3 and Amp4, which grew within the Amp2 aggregates or in Amp1-Amp2 or Cpx-Amp2 boundaries. They occur as individual grains or in the form of zonal grains. The core is Amp3-Wnc to Ktp, whereas the rim is Amp4-Gru. Some individual grains are Sdg in P17C (Figures 17B and 23).

We also found Amp3-type-Krs to Fkrs, or Prg in association with Cpx and Phl in troctolite P18A and dunite P18D. In a detail view, Amp replaces Cpx there (Figure 11F,H).

*Secondary Amp5*

The low-Al Amp5 (Tr, Act) aggregates (Figure 23) overgrow and partly to totally replace the Am1–4 generations and Cpx in mafics.

*Metamorphic Amp6*

Amphibolites from the inferred ophiolitic thrust sheet hanging walls (P12A,B, P10F) contain metamorphic Amp-Mhb to Prg (Figure 23). The sheared domains contain relics of porphyroclastic Mhb and Prg within recrystallized Mhb and Prg and the newly formed Tr-Act-Ep-Zo-Czo-Chl-Phg-Ab aggregates.

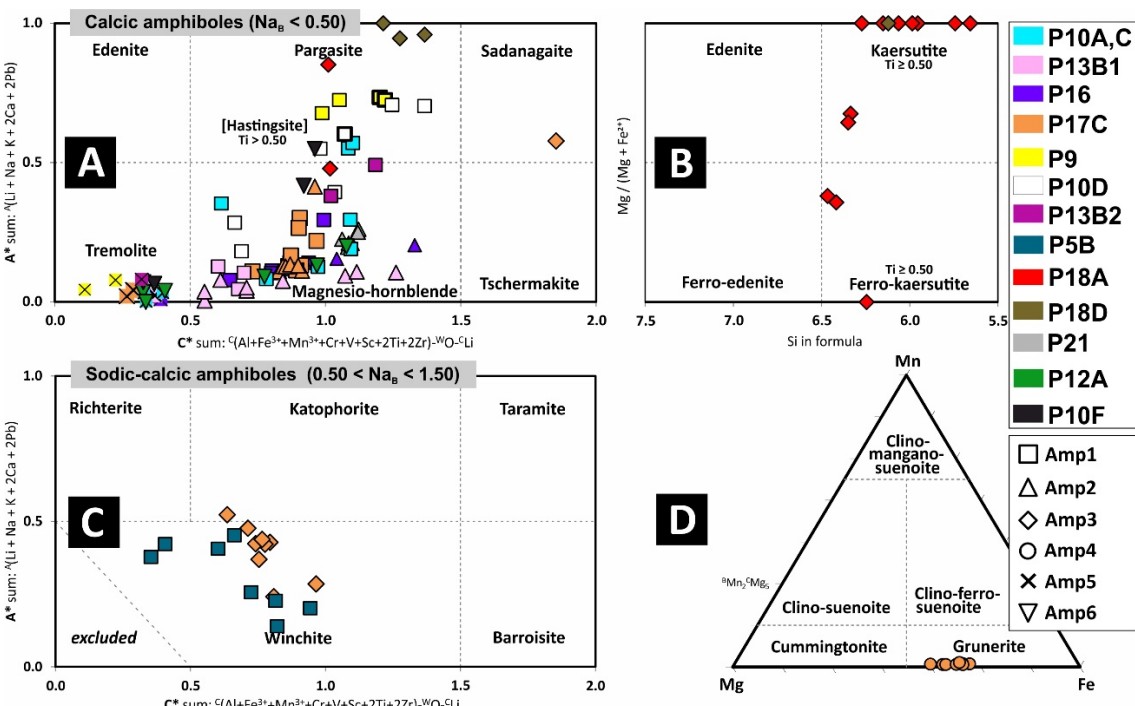

**Figure 23.** (**A–D)** Amphibole chemistry in ultramafic and mafic rocks. Classification from [56]. Amp1 with thick frame in (**A**) is Hst.

4.5.6. Plagioclase

Plagioclase of gabbros, dolerites, troctolites, and Pl peridotites has primary higher An content, which is only rarely preserved (e.g., P9). However, reactional and alteration processes decreased An and increased Ab components in Pl (Figure 24).

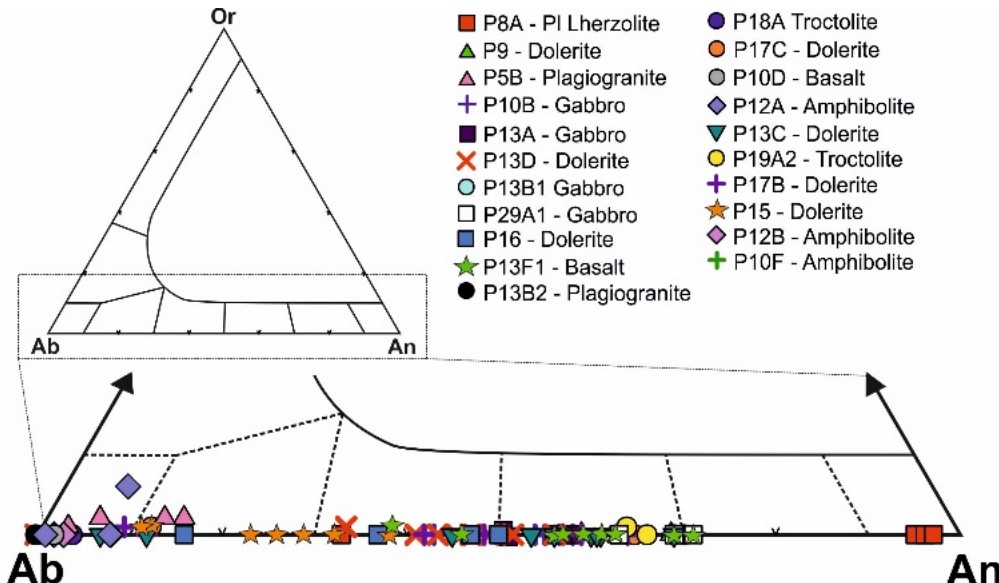

**Figure 24.** Plagioclase chemistry.

**Table 2.** Mineral chemistry of ultramafic rocks.

| Oxide (wt.%) | SiO$_2$ | TiO$_2$ | Al$_2$O$_3$ | Cr$_2$O$_3$ | FeO | MgO | CaO | Na$_2$O | NiO | Cr# | Mg# |
|---|---|---|---|---|---|---|---|---|---|---|---|
| **Lherzolite** | | | | | | **Samples P-2,6,7** | | | | | |
| Ol (early magmatic) | 40.12–40.97 | – | – | – | 8.84–9.91 | 48.97–50.97 | – | – | 0.33–0.44 | – | 90.02.-90-92 |
| Opx1 (early magmatic) | 51.58–56.17 | 0.00–0.20 | 1.82–5.93 | 0.52–0.93 | 5.90–6.86 | 31.73–34.50 | 0.49–1.24 | – | – | – | 89.19-91.12 |
| Opx2 (late magmatic) | 55.59–56.03 | 0.11-0.20 | 1.64-1.93 | 0.44-0.52 | 6.47-6.68 | 34.11-34.14 | 0.79-1.04 | – | – | – | 90.12-90.38 |
| Cpx1 (early magmatic) | 51.10–53.20 | 0.03–0.40 | 2.40–4.73 | 0.85–1.33 | 2.00–3.18 | 16.25–18.09 | 21.86–24.19 | 0.02–0.20 | – | – | – |
| Cpx2 (late magmatic) | 51.54–52.93 | 0.14–0.21 | 2.28–3.69 | 0.67–1.04 | 1.84–2.98 | 17.20–17.81 | 21.84–24.15 | 0.14–0.15 | – | – | – |
| Cpx3 (late magmatic, symplectitic) | 50.64–51.64 | 0.08–0.14 | 2.63–4.51 | 0.62–1.19 | 2.16–2.66 | 16.51–17.15 | 23.31–24.57 | 0.04–0.09 | – | – | – |
| Spl1 (early magmatic) | – | 0.01–0.58 | 20.50–43.54 | 24.32–39.49 | 14.19–29.50 | 7.89–16.42 | – | – | 0.14–0.21 | 45.72–51.72 | – |
| Spl2 (late magmatic) | – | 0.01–0.36 | 24.12–47.87 | 19.71–39.25 | 14.19–25.08 | 9.16–16.80 | – | – | 0.08–0.28 | 41.17–48.67 | – |
| Spl3 (late magmatic, symplectitic) | | 0.03–0.08 | 45.85–51.66 | 15.68–17.32 | 12.72–19.03 | 16.39–18.54 | | | 0.16–0.26 | 17.67-19.32 | |
| **Pl Lherzolite** | | | | | | **Sample P-8A** | | | | | |
| Ol (early magmatic) | 38.77–40.59 | – | – | – | 9.56–9.86 | 50.39–50.94 | – | – | 0.39–0.46 | – | 90.21-90.44 |
| Opx1 (early magmatic) | 54.67–55.51 | 0.06–0.08 | 2.95–3.81 | 0.82–1.07 | 6.52–6.70 | 33.67–34.10 | 1.02–1.16 | – | – | – | 90.06-90.20 |
| Opx2 (late magmatic) | 54.92–55.23 | 0.08–0.09 | 2.06–2.15 | 0.45–0.66 | 6.56–6.77 | 33.87–34.60 | 0.60–1.47 | – | – | – | 90.11-90.21 |
| Cpx1 (early magmatic) | 51.40–52.06 | 0.15–0.19 | 2.84–3.18 | 1.13–1.23 | 2.36–2.50 | 17.20–17.43 | 23.91–24.42 | 0.16–0.18 | – | – | – |
| Cpx2 (late magmatic) | 51.65–52.19 | 0.16–0.19 | 2.29–2.43 | 0.73–0.82 | 2.45–3.09 | 17.90–19.39 | 22.08–24.45 | 0.11–0.14 | – | – | – |
| Spl1 (early magmatic) | – | 0.14–0.30 | 25.20–38.26 | 26.36–38.44 | 21.66–24.94 | 10.52–13.24 | – | – | 0.10–0.18 | 47.66–48.93 | – |
| Spl2 (late magmatic) | – | 0.01–0.20 | 43.79–52.35 | 12.22–22.84 | 16.15–22.96 | 13.17–17.14 | – | – | 0.19–0.31 | 21.51-39.00 | – |
| Pl (from percolating melt) | 42.07–44.18 | – | 35.70–36.07 | – | – | – | 9.70–19.90 | 0.39–10.98 | – | – | – |
| **Harzburgite** | | | | | | **Samples P-11B,21,29A2,C2** | | | | | |
| Ol (early magmatic) | 39.43–40.75 | – | – | – | 8.78–12.01 | 48.33–50.67 | – | – | 0.24–0.42 | – | 84.51-91.02 |
| Opx1 (early magmatic) | 53.73–56.17 | 0.00–0.18 | 1.85–5.78 | 0.41–1.04 | 5.68–7.59 | 32.15–34.96 | 0.45–1.42 | – | – | – | 88.31-91.34 |
| Cpx1 (early magmatic) | 50.71–52.94 | 0.02–0.65 | 2.69–5.64 | 0.76–1.37 | 1.89–2.71 | 16.23–17.41 | 20.87–23.45 | 0.12–0.28 | – | – | – |
| Cpx2 (late magmatic) | 51.32–53.14 | 0.00–0.82 | 2.18–4.77 | 0.67–1.09 | 2.05–2.84 | 16.54–17.55 | 20.48–24.00 | 0.16–0.28 | – | – | – |
| Spl1 (early magmatic) | – | 0.00–1.61 | 14.57–30.61 | 35.55–42.47 | 18.86–33.45 | 5.44–13.05 | – | – | 0.09–0.20 | 47.85–56.09 | – |
| Spl2 (late magmatic) | – | 0.01–0.05 | 53.14–54.91 | 13.47–15.09 | 11.46–12.74 | 17.88–19.17 | – | – | 0.28–0.32 | 36.05-36.69 | – |
| Amp-P21 (from percolating melt) | 48.08–48.86 | 0.24–0.34 | 10.90–11.84 | 1.19–2.22 | 2.46–2.84 | 18.51–18.92 | 11.95–12.47 | 0.91–1.10 | – | – | – |
| **Dunite** | | | | | | **Samples P-11A1,18D** | | | | | |
| Ol (early magmatic) | 40.17–40.74 | – | – | – | 8.24–9.49 | 49.65–51.28 | – | – | 0.31–0.40 | – | 90.55–91.74 |
| Spl (early magmatic) | – | 0.12–1.17 | 12.13–26.67 | 37.07–53.21 | 19.88–25.21 | 7.97–14.79 | – | – | 0.00–0.25 | 48.68–56.38 | – |
| Cpx-P18D (from percolating melt, incl. Phl) | 49.51–51.20 | 0.95–1.25 | 3.84–4.98 | 1.44–1.65 | 2.58–2.72 | 14.81–15.76 | 22.61–23.64 | 0.80–1.20 | – | – | – |
| Amp-P18D (from percolating melt, incl. Phl) | 42.23–43.96 | 3.28–4.83 | 11.10–12.26 | 2.32–2.97 | 3.61–4.06 | 16.40–17.44 | 10.43–11.68 | 4.20–4.91 | – | – | – |

**Table 3.** Mineral chemistry of mafic rocks.

| Oxide (wt.%). | SiO$_2$ | TiO$_2$ | Al$_2$O$_3$ | Cr$_2$O$_3$ | FeO | MgO | CaO | Na$_2$O | NiO | Cr# | Mg# |
|---|---|---|---|---|---|---|---|---|---|---|---|
| **Troctolite** | | | | | | Samples P-18A,19A2 | | | | | |
| Ol (inherited from dunite) | 38.91–40.51 | – | – | – | 9.89–16.51 | 43.57–50.55 | – | – | 0.17–0.38 | – | 82.84-90.06 |
| Spl (inherited from dunite) | – | 1.34–2.25 | 18.96–25.67 | 30.30–37.71 | 20.76–39.50 | 5.58–13.40 | – | – | 0.09–0.27 | 41.38-56.41 | – |
| Pl (from percolating melt) | 50.34–51.78 | – | 30.34–31.91 | – | – | – | 12.60–14.50 | 3.44–3.89 | – | – | – |
| Cpx (from percolating melt) | 49.37–52.40 | 0.68–1.92 | 3.20–4.85 | 0.73–1.45 | 2.84–3.71 | 14.85–17.11 | 20.87–22.99 | 0.29–0.65 | – | – | – |
| Amp-P18A (from percolating melt, incl. Phl) | 40.55–46.03 | 3.48–5.47 | 11.09–16.94 | 1.39–3.39 | 1.52–2.92 | 16.77–25.68 | 0.11–11.65 | 0.75–6.29 | – | – | – |
| **Gabbro** | | | | | | Samples P-10A-C,13B1,26B,23A,29A1,29C1 | | | | | |
| Cpx (early magmatic) | 47.99–54.66 | 0.06–1.32 | 0.11–5.14 | 0.05–1.09 | 2.65–13.58 | 11.31–19.57 | 17.57–25.17 | 0.03–0.45 | – | – | – |
| Pl (magmatic+alteration) | 50.04–68.42 | – | 19.08–31.36 | – | – | – | 0.27–14.03 | 11.21–11.50 | – | – | – |
| Cpx(2) (after dynamic recrystallization) | 50.76–50.79 | 0.83–1.08 | 2.81–2.92 | 0.00–0.02 | 5.49–5.89 | 15.46–15.94 | 21.37–21.97 | 0.25–0.34 | – | – | – |
| Amp1 (magmatic) | 43.04–51.11 | 0.10–3.50 | 5.31–10.03 | 0.00–0.04 | 10.53–22.29 | 9.20–16.96 | 0.05–11.80 | 0.74–2.44 | – | – | – |
| Amp2 (late magmatic) | 44.91–51.77 | 0.07–3.07 | 5.02–11.60 | 0.00–0.33 | 8.52–17.86 | 10.33–17.11 | 9.67–11.66 | 0.16–2.58 | – | – | – |
| Amp5 (post-magmatic alteration) | 51.44–55.87 | 0.22–0.32 | 1.91–3.35 | 0.00–0.16 | 8.28–19.66 | 11.23–19.50 | 10.53–12.36 | 0.23–0.46 | – | – | – |
| **Plagiogranite** | | | | | | Sample P-5B | | | | | |
| Pl (magmatic+alteration) | 64.57–68.02 | – | 19.15–22.39 | – | – | – | 0.16–3.24 | 9.64–11.04 | – | – | – |
| Amp1 (magmatic) | 53.42–57.03 | 0.41–1.63 | 0.08–0.19 | 0.00–0.03 | 15.13–25.69 | 6.59–13.57 | 4.50–6.28 | 3.82–5.95 | – | – | – |
| **Dolerite** | | | | | | Samples P-11A1,18D | | | | | |
| Ol-P9 (early magmatic) | 36.78-37.61 | – | – | – | 30.81–31.50 | 32.38–32.93 | – | – | 0.01–0.05 | – | 64.96-65.62 |
| Opx-P9 (magmatic) | 53.69–55.04 | 0.04–0.05 | 0.70–1.27 | 0.00–0.06 | 17.78–18.45 | 26.36–26.87 | 0.40–0.42 | – | – | – | 71.91-73.02 |
| Cpx (magmatic) | 48.55–53.03 | 0.09–1.61 | 0.69–4.71 | 0.00–0.64 | 4.56–21.45 | 9.03–18.44 | 14.91–23.60 | 0.11–1.04 | – | – | – |
| Pl (magmatic+alteration) | 51.74–66.84 | – | 20.00–31.58 | – | – | – | 1.48–14.03 | 3.43–11.80 | – | – | – |
| Amp1 (magmatic) | 41.62–52.24 | 0.10–3.85 | 4.45–12.86 | 0.00–0.43 | 10.74–17.07 | 12.20–15.37 | 8.87–11.72 | 0.47–3.33 | – | – | – |
| Amp2 (late magmatic) | 45.25–53.88 | 0.09–1.12 | 4.86–11.62 | 0.00–0.07 | 11.02–17.70 | 11.00–17.20 | 9.65–11.34 | 0.38–2.24 | – | – | – |
| Amp3-P17C (from percolating melt/fluids) | 50.86–53.22 | 1.33–4.13 | 0.83–15.05 | 0.00 | 17.90–27.24 | 6.08–12.44 | 4.26–9.49 | 2.98–4.90 | – | – | – |
| Amp4-P17C (from percolating melt/fluids) | 51.67–53.68 | 0.17–1.14 | 0.25–1.03 | 0.00–0.02 | 28.02–33.94 | 8.15–12.32 | 0.83–4.97 | 0.24–0.73 | – | – | – |
| Amp5 (post-magmatic alteration) | 51.67–58.93 | 0.03–1.56 | 0.42–3.66 | 0.00–0.06 | 4.94–28.04 | 6.49–20.94 | 4.67–12.66 | 0.03–1.02 | – | – | – |
| **Basalt** | | | | | | Samples P-10D,13F | | | | | |
| Cpx-only in P13F (magmatic) | 47.13–48.50 | 1.26–1.97 | 3.11–6.29 | 0.00–0.30 | 7.00–15.15 | 9.11–13.36 | 19.80–20.99 | 0.27–0.38 | – | – | – |
| Pl (magmatic+alteration) | 52.69–68.10 | – | 19.79–31.34 | – | – | – | 0.41–13.87 | 3.11–11.37 | – | – | – |
| Amp1-only in P10D (magmatic) | 40.18–50.46 | 0.11–2.73 | 4.84–14.94 | 0.00–0.09 | 13.05–16.75 | 9.51–15.83 | 11.42–12.23 | 1.00–2.71 | – | – | – |

## 5. Discussion of Evolutional Stages of the OOC

Mantle peridotites exposed in the OOC are composed of lherzolites, harzburgites, and rare dunites. A part of the dunite layers in peridotites documents transition to troctolites due to secondary Pl and Cpx enrichment from percolating melts.

Gabbroic rocks (Cpx-Pl or Cpx-Pl-Amp) were determined from a borehole core material (drilled thickness of approximately 100 m), an ophiolitic breccia, and dykes in peridotites. Some massive medium- to fine-grained gabbro-dolerites (P15, P16) in contact with peridotites (Figure 2) may indicate the lowermost part of an inferred ophiolitic gabbroic layer, which rapidly cooled on an underlying serpentinized peridotite layer. There are common transitions to fine-grained basaltic layers in dolerite dykes (e.g., P17A–C). Until now, only one dolerite dyke (P9) crosscutting peridotite has a special Ol-Opx-Cpx-Amp-Pl mineral composition in comparison with the dolerites composed of Cpx, Pl, and Amp.

One larger intrusion of Qz-rich tonalitic plagiogranite (P5) is exposed in direct contact with peridotites (P4 and P6). On the other hand, a thin Pl-rich Cpx-Amp-bearing leucogabbro vein (P13B2) crosscuts a gabbro dm-size fragment (P13B1) from ophiolitic breccia. These rocks may have formed from strongly fractionated gabbroic melts or by a gabbro partial melting at the depth (e.g., [57]).

Rare dm-size basalt fragments and reddish radiolarites occur in ophiolitic breccia where, however, the other members of the OOC clearly predominate, such as serpentinites, gabbros, and dolerites. One basalt dyke (P10D), crosscutting layered gabbro (P10A), was registered in a borehole core profile.

This rock assemblage may indicate a thin gabbro layer remnant, whereas the sheeted dyke and basalt layers are practically missing in the OOC. In addition, the diachronous magmatism related to the extension regime is typical of slow-spreading ridge/transform systems, where the upper parts of the ophiolitic complexes are usually not developed (e.g., [13], and references therein). At this investigation stage, we cannot exclude that some basalts or gabbros from the ophiolitic mélange may have arc/back-arc character as in the KKOC [28,36], although we did not find signatures of the intraoceanic subduction.

Inferred Neogene volcanics in Figure 2 [33,45,46] were dated as Oligocene (30.40±1.31 to 28.52 ± 1.11 Ma) high-K calc-alkaline and shoshonite rocks. They occur within Cenozoic magmatic formation of Dinarides, which successively arose between 55 and 29 Ma and are genetically related to the collision of Apulia (Africa) and Tisia (Euroasia) [58,59].

### 5.1. Spinel Peridotites and Dunite Layers

The Ol-Opx1-Cpx1-Spl1 boundaries in peridotites of the OOC are infilled with the melt-crystallized undeformed medium-grained Cpx2 (less Opx2) and Spl2 aggregates and fine-grained Cpx3-Spl3 symplectites. We suggest an alternative model to the model of percolating reactive Ol-rich melt through the pre-existing peridotites [2–7,16]. All the observable signatures—the HT-deformed Px1 porphyroclasts—and Spl1 partial dissolution may have occurred during the final crystallization in subsolidus stage. These porphyroclasts and coarse-grained Spl1 might have reacted with the rest grain boundary melt by their dissolution. The irregular shape of the deformed Px porphyroclasts and Spl1 is evident, and the grain boundaries are infilled by the late-magmatic Cpx2-Opx2-Spl2 aggregates or locally by the Cpx3-Spl3 symplectites, with the latter indicating a final rapid crystallization stage of the rest melt in the boundaries of older phases. We suppose that most of the Al from the inferred rest melt was consumed by Spl3 (with the highest Al content compared to Spl1, 2; Table 2) intergrowing with mostly Al-poor Cpx3. A pre-existing Grt dissolution as an additional Al source cannot be excluded. The increased Ni content in Spl2 and 3 may suggest dissolution of Ol in the late magmatic to subsolidus stage.

On the other hand, dunites may form by the reactive dissolution when Ol-normative basaltic melts percolate through a harzburgite or lherzolite matrix [3,7,60–62]. However, such a new melt influx is not clearly observable in the reaction domains. Then, considering

the dunite formation, we could infer the channels of a complete pyroxene-rich (basaltic) melt extraction from partially molten peridotite forming the Ol-Spl dunite layers, with the latter providing permeable domains for a percolating melt and the formation of Pl peridotites and troctolites.

Lherzolites, harzburgites, and dunites of the OOC may have thus formed at a deeper lithospheric mantle environment in the Spl stability field.

### 5.2. Melt Impregnation of Peridotites

The formation of troctolites in the Donja Paklenica Valley (P18A) and P19A2 in the north (Figure 2) suggests a melt-rock interaction-dominated process, which involved Ol dissolution and crystallization of interstitial Pl and minor Cpx. This stage might have occurred from transitional Spl-Pl facies depths according to Spl partial replacement by Pl. During this process, Spl harzburgites might have evolved to impregnated Pl peridotites at estimated depths of ca. 30 to 10 km (~1 to 0.3 GPa).

The presence of newly formed Cpx without or with Amp in troctolites may indicate (1) a dry melt percolation related to the Cpx-Pl gabbro dyke formation in a close neighbourhood (P29A1, P29C1), or (2) a hydrated melt percolation related to the Cpx-Pl-Amp dolerite dykes (P17A–C in a close neighbourhood) and plagiogranite formation.

Harzburgite P21 exclusively contains Mhb after magmatic Cpx and Opx and their pyroxene exsolution lamellae, which may have also formed via the interaction of harzburgite with percolating melts and fluids, such as Pl peridotites (e.g., P8A).

Some Cpx-Pl gabbroic dykes (P19A1, P29A1, P29C1) contain assimilated Ol xenocrysts from hosting harzburgite close to the melt–peridotite interface. Such Ol is enriched in iron (Figure 19) due to its proximity to the percolating basaltic/gabbroic melt, and such a reaction was tested by [11].

The impregnation by pervasive melt flow and the formation of Pl-peridotites and troctolites bound to the dunite layers confirms that the dunitic peridotites are the most permeable environment for the pervasive percolating melts. There, relatively high volumes of melt could infiltrate the peridotites by porous flow, producing troctolites. The replacive character of troctolites in the dunitic mantle protoliths and the compositional evolution of the percolating melt during melt–rock interactions reported [13] from the Erro-Tobbio (Italy) impregnated mantle peridotites, which are similarly associated with a troctolitic body and crosscutting gabbroic dykes, as in the OOC.

Migrating melt was likely produced by mantle upwelling and melting related to extension and the thinning of thermal lithosphere, which is typical of slow-spreading oceanic lithosphere settings (e.g., [4]). The Pl-peridotites coexisting with scanty oceanic crust corroborate the model of a cold mantle belt, which is, e.g., known from the melt-poor (ultra) slow equatorial Atlantic Ocean Romanche transverse ridge [63]. The melt retention at the base of such a conductively cooled boundary layer (the thermal lithosphere) lowers the density of the upper mantle, thus driving the upwelling of the mantle [21,64]. The percolating melt was likely formed by decompression and partial melting of a refractory peridotite in the asthenosphere and via a reaction of this melt with peridotite at the base of the thinning thermal lithosphere [4,65].

### 5.3. Estimated Magmatic Crystallization P–T Conditions from Peridotites

The late magmatic to subsolidus temperatures of ca. 1000–850 °C, prior to melt impregnation, were approximated from lherzolite P2 and harzburgites P11B and P21 using Opx1 and Cpx1 for the two-pyroxene thermometry [48]. Ca-in-Opx1 thermometry of [48] provided T of 1028–1068 °C, whereas Ca-in-Opx2 thermometry gave a lower T of 909–961 °C. Estimated P values of 1.1–0.9 GPa are consistent with a lower P limit for the Spl stability field.

For comparison, the thermometric estimates for peridotite from the mid-Atlantic Romanche transverse slow-spreading ridge are in the range of 750–1050 °C. The wider temperature interval may reflect localized reheating of relatively cold peridotites by migrating melts [63]. Similarly, temperatures calculated from average Opx compositions with the Ca-in-Opx thermometer of [48] are higher for Pl peridotites, up to 1200 °C, and lowest for spinel peridotites, approximately 900 °C, in the KKOC. This may indicate overheating of Spl peridotites by hot percolating melts [28].

### 5.4. Estimated Magmatic Crystallization P-T Conditions from Troctolites, other Gabbros, and Dolerites

Single Cpx thermobarometry [49] from troctolites (P18A) and gabbro-dolerites (P9, P17C) of the OOC constrained magmatic crystallization T between 1200 and 1100 °C at 0.45–0.15 GPa at the estimated depth of 13–5 km. Similar estimated P–T conditions may suggest a narrow time interval between the formation of troctolites, gabbroic-doleritic dykes, and Pl peridotites.

Amphibole2 (Mhb) aggregates at the rims of Amp1 and in grain boundaries of the Cpx, Amp1, and Pl terminated magmatic crystallization of gabbros and dolerites via the formation of prismatic to needle-like Amp2 aggregates of the same chemical composition as the Amp1 (Figure 23; Table 3). These transitional textures from coarse-grained Amp1 to medium-grained prismatic, up to needle-like Amp2 aggregates, may indicate an accelerated cooling following the gabbroic melt intrusion in cooler peridotite.

Troctolite P18A contains newly formed Amp (replacing Cpx of troctolite) classified as Mhb to Prg, or Na-Ti-rich Krs to Fkrs, the source of which could be hot hydrated percolating melts. Similarly, secondary Amp3 Na-rich Wnc to Ktp of dolerite dyke P17C is chemically close to Amp1-Wnc of a plagiogranitic intrusion (P5); the latter could be a source of these Na-rich Amp3.

The change from a pervasive to channel style of the melt–rock interaction indicates conductive cooling of these peridotites and the emplacement of gabbroic and doleritic dykes into extensional fractures of the less ductile upper part of the thermal lithosphere layer at ca. 15–5 km, before their exposure to the ocean-floor metamorphic alterations. Some Cpx-Pl dykes (P23A) show mylonitic banded structures and dynamic recrystallization of Cpx layers indicating T of at least 700 °C. These T values may indicate a temporary continuation of the extensional ductile regime after the dyke emplacement into cooler peridotites.

The melt impregnation modes are interpreted to have accompanied progressive tectonic uplift from asthenospheric levels across the ductile/brittle transition up to shallower depths. For example, barometric estimates from the Romanche peridotites suggest that they were impregnated at minimum depths of ca. 9–12 km [61], while the maximum depth interval of 9–12 km was estimated for magmatic impregnation in the form of dykes and veins for the mantle section of the Othris ophiolites in the range of 1200–1000 °C [65], which is similar to the OOC.

### 5.5. Ocean Floor Alterations

The ocean-floor alterations of the OOC include serpentinization and rodingitization of the whole sequence of the OOC. Typical hydration products of dykes are Tr, Act, H-Grs, and Chl at the expense of Cpx and Amp1, or albitization (spilitization) of Pl, Ep, and Czo. Peridotites underwent serpentinization and rodingitization at very low T and P. Previous authors [66] suggested 200–300 °C at maximum 0.05 GPa for gabbro rodingitization by fluids related to serpentinization of host peridotites.

### 5.6. Accretionary Wedge Thrust-Sheet Hanging Wall Amphibolites

Metamorphic Amp Mhb to Prg and Pl from amphibolite P12A suggest T 620 °C and P 0.85 GPa (Ti-in-Amp thermometry and Amp-Pl thermobarometry of [50,51]), which we

interpreted as an ophiolite thrust sheet hanging wall at a depth of ca 28 km. These conditions are most likely related to an early exhumation stage of this ophiolitic thrust sheet in an accretionary wedge, the formation of which followed the obduction.

Amphibolite P12B from an exhumation shear zone yielded 300 °C at 0.5 GPa by a combination of Chl thermometry and Si-in-Phg barometry. The recrystallization aggregates of Mhb and Prg occur within newly formed Chl, Act, Tr, Phg, Ab, Czo, Zo, Rt, and Ttn aggregates crosscut by Ap-Zrn veinlets, which may represent the lower-T amphibolite to greenschist facies exhumation conditions.

Petrological study of the Ozren Massif by [33] provided the first data on amphibolites. They reported the presence of normal amphibolites, rare pyroxene amphibolites, and amphibole-zoisite schists. These rocks mostly occur at the contact of ultramafics with Jurassic volcanic–sedimentary formation (mélange). Amphibolites are also stratified, and their foliation and inner structures coincide with the layering of ultramafics. These relationships indicate that amphibolites and ultramafics represent members of a uniform geological complex. Accordingly, amphibolites are orto-metamorphic rocks [33].

## 6. Conclusions

The investigated Ozren OC of the Neotethys Jurassic Dinaride Ophiolite Belt shows polystage mantle evolution recorded in peridotites, plagioclase peridotites, plagiogranites, troctolites and other gabbros, and gabbroic, doleritic, and fewer basaltic dykes.

The subsolidus magmatic stage of lherzolites and harzburgites is inferred from the deformed Opx1 and Cpx1 porphyroclasts and coarse-grained Spl1. The irregular shape of these partly dissolved phases seems to be a result of their interaction with the rest grain boundary melt, with the latter crystallizing the late-magmatic undeformed aggregates of Cpx2, Opx2, Spl2, and often Cpx3-Spl3 symplectites.

The suggested evolutionary trend is consistent with the determined mineral chemistry changes such as the distinct decrease in Cr# and $TiO_2$, and the $Al_2O_3$ increase, from Spl1 to Spl2 and 3 in ultramafics. By contrast, the $Al_2O_3$ decrease is characteristic of the early- to late-magmatic pyroxenes. Similarly, an increase in Ni in Spl2 and 3 may indicate Ol dissolution during the late-magmatic stage.

Troctolites exclusively follow rare dunite layers in peridotites. They display interstitial crystallization of Pl and less Cpx in the Ol matrix, thus providing evidence of dunite replacement by impregnating Pl-Cpx melt. Plagioclase-crystallizing melt, interacting with peridotite, was related to the cooling and crystallization of inferred fractionating melts within peridotites at the shallower lithospheric mantle level in comparison with the formation of Spl peridotites to dunites. This process is recorded in an iron increase in Ol of peridotites close to their interface with pervasive basaltic/gabbroic melt percolation through dunite layers, or focused melt percolation in gabbroic and doleritic dyke systems. The low-Na Cpx-Pl-Ilm-±Amp gabbroic and less basaltic dykes in peridotites aptly trace the inferred subridge extension and the relative uplift of peridotites due to mantle thinning.

The generation of Na-(Ti)-rich Amp (Wnc to Ktp, or Krs to Fkrs) associated with Phl or Bt occurs as primary magmatic in plagiogranites (P5), but as the secondary post-magmatic in troctolites (P18A), dunites (P18D), and some gabbro-dolerites (P17C). This Amp generation and Phl replace Cpx of troctolites, and this successive mineralization may indicate two types of percolating melt. The "dry" melt percolation can be related to the Cpx-Pl gabbro dyke formation, whereas the "hydrated" melt percolation produced the Cpx-Pl-Amp gabbro-dolerite dykes and plagiogranites. The resulting products of peridotites and percolating melt interaction seem to also be Pl peridotites.

The ultramafic and mafic rocks underwent ocean-floor hydration (serpentinization, rodingitization, chloritization) at very low temperatures and pressure. The alteration of mineral products mainly occurs for minerals of the Srp group, Chl, Act, Tr, H-Grs, Ab, Ep, Czo, and Ttn.

The T of the magmatic to subsolidus stage of peridotites was estimated from Cpx1 and Opx1 to 1000–850 °C in the Spl stability field by the two-pyroxene thermometry. Ca-in-Opx1 thermometry provided T of 1028–1068 °C, whereas Ca-in-Opx2 thermometry gave a lower T of 909–961 °C at an estimated P of 1.1–0.9 GPa. The magmatic crystallization T of the gabbros, dolerites, and dykes was constrained to 1200–1100 °C at P 0.45–0.15 GPa by single Cpx thermobarometry. These estimates document that peridotites formed at the deeper mantle environment compared to Pl peridotites, troctolites, and gabbroic and basaltic dykes.

Amphibolites from the inferred ophiolitic thrust-sheet hanging walls provided approximate starting exhumation conditions of 620–600 °C and 0.85–0.6 GPa in an accretionary wedge by Ti-in-Amp thermometry and Amp–Pl thermobarometry. Exhumation shear zones with Act, Tr, Phg, and Chl yielded 300 °C at 0.5 GPa via the combination of Chl thermometry and Si-in-Phg barometry.

The Ozren OC suggests an environment of a slow-spreading ridge with a reduced gabbroic layer and practically missing sheeted dyke and basalt layers. It does not show a metamorphic overprint, excluding the obduction-related accretionary wedge ophiolitic thrust-sheet hanging walls metamorphosed in amphibolite facies.

**Author Contributions:** Conceptualization, M.P. and E.B.; methodology, M.P, O.N., S.U., P.R., and S.K.; software, O.N. and S.U.; validation, M.P., E.B., and F.K.; formal analysis, S.K.; investigation, M.P., S.U., E.B., F.K., O.N., P.R., and P.K.; resources, M.P., E.B., and P.K.; data curation, M.P., O.N., and S.U.; writing—original draft preparation, M.P., O.N., and S.U.; writing—review and editing, M.P., O.N., S.U., E.B., and F.K.; visualization, O.N. and S.U.; supervision, M.P.; project administration, M.P.; funding acquisition, M.P. All authors have read and agreed to the published version of the manuscript.

**Funding:** This research and the APC were funded by The Slovak Research and Development Agency Project No. APVV-19-0065 (M.P.).

**Acknowledgments:** The authors greatly acknowledge the suggestions of the three reviewers to improve the original manuscript.

**Conflicts of Interest:** The authors declare no conflict of interest.

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
