# Peer review of "Mineralogical-Petrographical Record of Melt-Rock Interaction and P–T Estimates from the Ozren Massif Ophiolites (Bosnia and Herzegovina)"

_minerals, doi:10.3390/min12091108_

Round 1

Reviewer 1 Report

The paper is written well. The authors made excellent efforts to address the issue. My main concern is the discussion. The authors included pertinent information in the manuscript. They attempted to explain it using the published domain, but it is only applicable on a local scale. The data should be justified and correlated on a regional scale, and its applicability should be addressed in the conclusion in a proper manner.

Reviewer 2 Report

The manuscript “Mineralogical–petrographical record of melt–rock interaction 2 and P–T estimates from the Ozren Massif ophiolites (Bosnia and Herzegovina)” by Marián Putiš, Ondrej Nemec, Samir Ustalić, Elvir Babajić, Peter Ružička, Friedrich Koller, Sergiy Kurylo and Petar Katanić is a valuable contribution to the knowledge of the mineralogical and petrographical characteristics of the Ozren Massif ophiolites. The authors realize a credible scenario for the Ozren Massif Neotethys Jurassic Dinaride Ophiolites by demonstrating their polystage mantle evolution in accordance with the geotectonic development. I advise publishing the present manuscript with minor correction, as suggested below.

Also since on the geological map (Fig. 2) there are present andesites and dacites, even they have been not studied, I may suggest including a short discussion on these rocks according to the current literature.

Suggestion for corrections:

line 21: use clinopyroxene;

lines 29, 309, 620: Interstitial not intersticial;

Line 52: It is desirable to explain the used symbols before mentioning: So, use: Olivine (Ol)-saturated:

lines 56, 58: as above: Plagioclase (Pl) rich; Pyroxene(Px) crystallization;

line 65: as above: Spinel (Spl)-facies;

line 66: as above: clinopyroxene (Cpx) and orthopyroxene (Opx);

line 87: as above: Cpx–(±Amphibole-Amp, Phlogopite-Phl);

line 98: explain Pl;

line 99: explain EPMA (electron probe micro-analysis);

lines 102-110: Add Magnetite- Mag to abbreviations;

line 259: explain Mhb and add to abbreviations.

Reviewer 3 Report

The paper provides a detailed, elaborated, well-illustrated and commendable description of the rock types and the component minerals.

It proposes a two-stage percolation of reactive peridotite by replacive dunite. The first stage of the transformation is not supported by the evidence invoked, namely strained pyroxene dissolution and neoformation of a second-generation pyroxene. Neoformed olivine is not widespread, and the explanation given is the evolution of the percolating magma towards a pyroxene saturated composition with no explanation of how and why such a process should occur – that means no evidence whatsoever of a replacive dunite. As stated in r78-79, the question arises whether the neoformed “boundary aggregates” formed from a residual melt or represent the product of a new melt influx. The paper fails to provide compelling evidence for the latter variant.

On the other hand, the second claimed stage (which then would be the first), leading to dunite and troctolite formation from the pre-existing peridotite by a melt influx is tenable.

The paper fails to explain also the origin of the hydrated melt responsible for late amphibole formation.

Specific comments are listed below:

r20 and following: I’d rather reserve the use of ”less” for singular, uncountable and plurar-deffective nouns and use ”fewer” or an equivalent, considering also the ambiguity induced by an adjective immediatly following – less basaltic...

r25-26: unclearly explained relationship between the olivine-rich melt dissolving pyroxene and the pyroxene-saturated melt dissolving olivine

r30 interstitial; and as above, although less phlogopite sounds right

r32: again in ”less basaltic dykes”, which supposedly are not less basaltic, but fewer.

r33: ”subridge extension and mantle exhumation towards the ocean floor” sounds preternatural, rephrasing should be useful in understanding what is means

r40: no way a hanging wall be a sole

r43: either singular or plural, we can’t have them both

r36-44: a geotectonic interpretation/assessment of the PT-data would be useful

r52-53: at temperatures preventing crystallization

r102-110 Mhb, Par, Wnc, Ktp abbreviations used in text not explained, though listed in [22]. It is a good idea to set down the abbreviations instead of sending the reader to browse a still insufficiently assimilated reference

r118 overlying or something, not overstepping

r119 of supra-subduction

r138 troctolite is also a gabbro

r129 units mentioned in the text should be identifiable on the map (Eastern Vardar, Sava Zone, aso.)

r130 please correct the erroneous caption for Fig. 1

r136 from the map Krivaja-Konjuh is not exactly neighbouring the Ozren complex

r139 and fewer basalts

r145-146 high Na/Na2O in Cpx

r147 have significantly lower Na in Cpx

r149 Al2O3 in Cpx as low as 2% and TiO2 below 0.1%

r171-172 Fig. 3 is superfluous since the sample numbers appear already in Fig. 2

r175 for constraining

r176 provide a database for future geochemical …. studies.

r181 on polished thin sections

r186 (more than 600 analyses) is good, but not especially relevant

r186-187 the voltage is not accelerated; it accelerates the electrons of the beam and is called therefore accelerating potential

r187 a beam 3-5 μm wide is in fact slightly defocused, not focused

r220 bent

r262 why already?

r264 eastern

r294-295 why moreover?

r305 why quotation marks?

r309 interstitial

r318 unclear what is meant by “inner coronas”

r320 comma misplaced; not necessary anyway

r342 lacks quartz – why is it then a plagiogranite?

r343 in a Pl-dominated matrix

r383-384 reference to Fig, 15 A, B should be made immediately after ”undeformed” – the pictures don’t illustrate ophitic textures and accessory minerals

r412-413 in the pictures do not illustrate clear ophitic textures, one cannot distinguish oiko- and chadacrysts

r423-424 it is difficult to reconcile Amp2 associated with chlorite (Fig. 17C) or filling (pseudomorphing) subhedral contours (Fig. 17B) with the claimed late-magmatic character

r 432 it likes?

r441 again a sole in the hanging wall

r489, 498-499 at least mineral names should be given in the singular form – enstatite, diopside and augite

r512 the chemical composition of successive Amp generations separated from microtextures – or something similar

r550 ingrow?

r558 a distinction between calcic and sodic-calcic amphibole is required in order to understand the difference between Fig. 23A and Fig. 23C. This is very difficult if no badly missed structural formulae are given, because the wt% oxide analyses are not very helpful.

R599 apparently the cited papers place the reactive peridotites in the lithospheric mantle, not in the asthenosphere

R600-602 difficult to understand, is there something missing?

R603 reactive rest melt?

r605 an explanation should be given with respect of these “signatures of reactive peridotites”

r624 it is strange to invoke a dry melt when amphibole is formed. Dykes are obviously emplaced later than the supposed melt percolation in their host rocks.

R729 again less basaltic

R again soles termed as hanging walls

R787 “Please turn to the for the term explanation.” should be deleted

Round 2

Reviewer 3 Report

I am satisfied with the revised version and the way the comments were handled. I feel two of the comments were not addressed possibly because  a lack of clarity their formulation. By Na/Na2O I did not mean the actual ratio, which we all know is a constant, but two alternative ways to write Na contents (as in and/or). Fine with Na expressed as Na2O, so "the high Na2O contents in Cpx.." @ r145-146, now r149.

Fig. 23 A and C still need an addition in order to make the reader clearly understand that it is about two parallel sections in the amphibole compositional space. So "calcic amphiboles" and "sodic-calcic amphiboles", or NaM4, NaB <0.5 and 0.5<NaB<1.5, or Ca >1.5 and 0.5<Ca<1.5 should be added to the two figures in order to distinguish them.

The diablastic intergrowth in Fig 5f is very interesting because it could not have formed from the reaction of the adjacent phases, both poor in alumina, nor is the local presence of such an Al-rich melt likely. Similar textures result from the breakdown of garnet - but this is a comment with no hints or intended consequences.
